# Polymorphism in Glu-Phe-Asp Proteinoids

**DOI:** 10.3390/biomimetics10060360

**Published:** 2025-06-03

**Authors:** Panagiotis Mougkogiannis, Andrew Adamatzky

**Affiliations:** Unconventional Computing Laboratory, University of the West of England, Bristol BS16 1QY, UK; andrew.adamatzky@uwe.ac.uk

**Keywords:** thermal proteins, proteinoids, microspheres, unconventional computing

## Abstract

Glu-Phe-Asp (GFD) proteinoids represent a class of synthetic polypeptides capable of self-assembling into microspheres, fibres, or combinations thereof, with morphology dramatically influencing their electrical properties. Extended recordings and detailed waveforms demonstrate that microspheres generate rapid, nerve-like spikes, while fibres exhibit consistent and gradual variations in voltage. Mixed networks integrate multiple components to achieve a balanced output. Electrochemical measurements show clear differences. Microspheres have a low capacitance of 1.926±5.735μF. They show high impedance at 6646.282±178.664 Ohm. Their resistance is low, measuring 15,830.739 ± 652.514 mΩ. This structure allows for quick ionic transport, leading to spiking behaviour. Fibres show high capacitance (9.912±0.171
μF) and low impedance (209.400±0.286 Ohm). They also have high resistance (163,067.613 ± 9253.064 mΩ). This combination helps with charge storage and slow potential changes. The 50:50 mixture shows middle values for all parameters. This confirms that hybrid electrical properties have emerged. The differences come from basic structural changes. Microspheres trap ions in small, round spaces. This allows for quick release. In contrast, fibers spread ions along their length. This leads to slower wave propagation. In mixed systems, diverse voltage zones emerge, suggesting cooperative dynamics between morphologies. This electrical polymorphism in simple proteinoid systems may explain complexity in biological systems. This study shows that structural polymorphism in GFD proteinoids affects their electrical properties. This finding is significant for biomimetic computing and sheds light on prebiotic information-processing systems.

## 1. Introduction

Proteinoids are synthetic polypeptides made by heating amino acids [1,2,3,4]. They give us a fascinating look at how life might have started and show promise for creating bio-inspired materials [5,6,7,8,9]. Sidney Fox first explored these self-assembling structures in the 1950s [1]. They mimic early protocells [10]. These structures can organize into complex forms [11]. They also show basic functions, like catalysis and membrane formation [12]. Glu-Phe-Asp (GFD) proteinoids are special. They can take on different shapes: tiny microspheres, long fibers, or a mix of both [13]. Each shape brings unique electrical behaviours. This polymorphism is the ability to exist in different forms. It is not a chemistry irregularity; it helps us understand how simple molecules create complex, life-like traits [14,15,16,17]. These include quick, nerve-like signals and stable memory-like patterns [18]. This paper will discuss various GFD proteinoid assemblies. It will measure their unique electrical signatures. Also, it will link structure to function by connecting shape to electrical behavior in these self-assembling systems. This paper is organized as follows: First, we look at how different GFD shapes behave electrically over long-term recordings. Next, we analyze the microscale structure of these assemblies using high-resolution electron microscopy. We measure the electrochemical properties of each morphology. Then, we link structure to function.

Structural polymorphism is not seen in proteinoid systems. It is a key feature found in many protein classes (Table 1). Proteins, like amyloidogenic ones, have structural diversity. This allows them to change between soluble and fibrillar states. This shift enables many functions in biological systems. This also includes the allosteric changes in hemoglobin.

The electrical activity of GFD proteinoids relies on their shape. Microspheres are tiny, round structures. They act like small power stations. As ions move across their surfaces, they create quick bursts of voltage [28,29]. Fibers stretch out like threads. They handle ions slowly, spreading them along their length. This creates smooth and lasting voltage shifts. Mixing the two blends these traits, balancing bursts with steadiness in a way that hints at cooperative networks [30]. These differences are not random. They show how structure affects function; it is like how a cell’s shape impacts its role in a living organism [31]. Studying GFD assemblies helps us see how polymorphism might have led to the shift from non-living chemistry to active, responsive systems in the prebiotic world [32]. This research also leads to new materials that mimic biological circuits [33]. This paper explores the electrophysiological diversity of GFD proteinoids. It highlights how microspheres, fibers, and their mixtures change over time. We study long recordings and detailed waveforms to uncover how electrical signatures work. This includes spikes in microspheres, gradual waves in fibers, and a mix of stability in networks. We want to find out how different structures affect potential. This will give us clues about the ancient origins of life and the future of bio-inspired tech [5]. GFD proteinoids highlight polymorphism. They connect simplicity and complexity. We will explore this theme in the upcoming sections.

Our study presents significant advances beyond the existing literature in several key aspects. Previous research has shown that proteinoids can form different structures and have electrical activity [12,34,35]. Yet, our work is the first to link specific shapes with unique electrical signatures in GFD systems. We go beyond earlier studies that looked at structure or electrical activity alone [36]. Our analysis links structure to function in a new way. We show how microsphere compartmentalization allows for action potential-like spiking. Also, the fibrillar architecture helps maintain stable potentials, like memory. This work leads to the quantitative electrochemical study of various proteinoid structures. It sets up key profiles for impedance, capacitance, and resistance. These profiles help explain the different electrical behaviors observed. Our mixed-morphology systems show new properties that we cannot predict from the individual parts. This points to fresh design ideas for biomimetic computing substrates. These substrates could have quick signal processing and reliable memory. These findings greatly improve the field. They show how simple biomolecular assemblies can process complex information. This opens new paths for neuromorphic computing materials that connect biological and artificial systems.

## 2. Methods and Materials

### 2.1. Materials

All amino acids, like L-glutamic acid, L-aspartic acid, and L-phenylalanine, were sourced from Sigma-Aldrich, Merck KGaA, Darmstadt, Germany. They were used as is, without extra purification. All other chemicals were of analytical grade.

### 2.2. Preparation of Glu-Phe-Asp Microspheres

Glu-Phe-Asp (GFD) proteinoid microspheres were prepared using a thermal condensation method. Equal masses of L-glutamic acid, L-aspartic acid, and L-phenylalanine were combined to yield a total mass of 5 g. The amino acid mixture was heated to 200 °C, which is above the boiling point of the amino acids. This was carried out under reflux conditions in a three-necked round-bottomed flask. It had a mechanical stirrer and a reflux condenser. The reaction stayed at this temperature for 3 h. This ensured complete polymerization and the formation of proteinoid material. After it cooled to room temperature (25 °C), the glassy material was mixed with hot distilled water (80–90 °C). Continuous stirring helped form microspheres through self-assembly. The suspension was allowed to cool gradually to room temperature and then stored at 4 °C for 24 h to stabilize the microsphere structures. The microsphere suspension was subsequently lyophilized using a freeze-dryer (operating at −50 °C and 0.01 mbar) for 48 h. The lyophilized material was filtered using membrane filters with a 0.45
μm pore size. It was then collected as a dry powder for further analysis and experimentation.

We have different protocols for microspheres and fibrillar structures. These protocols change the polymerization rate. This helps us promote different self-assembly pathways. In microsphere formation, heating all three amino acids (L-Glu, L-Asp, L-Phe) together in equal amounts forms uniform nucleation centers. This process balances hydrophilic and hydrophobic areas. This balance causes isotropic growth. The hydrophobic phenylalanine residues move inward, while the charged glutamic and aspartic acid residues point outward. This arrangement reduces interfacial energy, following the Gibbs–Thomson effect [37,38]. Our fibrillar preparation uses sequential polymerization. First, melting L-glutamic acid creates templates. These templates guide the later addition of L-aspartic acid and L-phenylalanine. This setup helps growth conditions that favor linear extension over spherical aggregation. Longer reaction times (4 h instead of 3) boost chain elongation. This means the growth of existing chains is promoted rather than starting new assembly centers. Reaction order and stoichiometry greatly affect supramolecular structure. Temperature control is very important. Keeping a steady 200 °C stops secondary nucleation events. This helps the primary fibrillar growth to stay on track. We optimized our cooling rates to keep these unique shapes. If we cool fibrillar preparations too quickly, they can break apart. However, cooling microsphere suspensions slowly helps to avoid clumping.

We have carried out FT-IR analysis of our GFD proteinoids before [39]. This confirmed that key peptide bonds and functional groups are present, showing the typical structures of proteinoids. The FT-IR spectra showed clear peaks at 1635 ${\mathrm{cm}}^{-1}$, which is the amide II band from peptide bond vibrations. It also displayed a peak at 1943 ${\mathrm{cm}}^{-1}$ for the amide I band from peptide group stretching. Additional signals appeared at 2108, 2349, and 3258 ${\mathrm{cm}}^{-1}$, linking to different functional groups found in amino acids. These spectroscopic results clearly showed that thermal condensation polymerization worked. They also confirmed the peptide backbone structure, which is key for the electrochemical properties discussed in this study. Other techniques like Raman or NMR could add useful structural details later. Our current FT-IR analysis, SEM morphology, and detailed electrochemical tests support the structure–function relationships in our manuscript.

### 2.3. Preparation of Glu-Phe-Asp Fibrillar Structures

The fibrillar GFD structures were prepared using a sequential thermal polymerization approach. Initially, 5 g of L-glutamic acid was heated under reflux conditions until complete melting occurred. To the molten mass, we gradually added a mix of L-aspartic acid and L-phenylalanine, totaling 10 g. We kept the temperature at 200 °C. This method helped create fibrillar shapes instead of spherical ones. The thermal condensation reaction lasted for 4 h. We stirred constantly to ensure the polymerization was uniform. The steps for cooling, processing, lyophilization, and filtration used the same protocol as microsphere preparation.

### 2.4. Preparation of Mixed Microsphere–Fiber Assemblies

We made heterogeneous assemblies using microspheres and fibers. First, we mixed pre-formed microspheres and fibers. The ratio was 50:50 (*v*/*v*) in an aqueous suspension. The mixture was gently sonicated for 30 s at 40% amplitude. This helped mix it well while keeping both components intact. The resulting suspension was then lyophilized and stored as described above.

### 2.5. Scanning Electron Microscopy (SEM)

We used a high-resolution scanning electron microscope, the FEI Quanta 250 FEG, originally developed by FEI Company, Eindhoven, The Netherlands, to study morphology. It ran at accelerating voltages of 2 to 5 kV. Samples were prepared by depositing a small amount of the proteinoid powder onto carbon tape mounted on aluminum stubs. We coated samples with a 10 nm thick layer of gold. This improved conductivity and image quality. We looked at several areas for each sample. This helped us to capture the main shapes accurately.

### 2.6. Electrical Recording Setup

We recorded extracellular potentials using a multi-electrode setup with Pt/Ir (platinum/iridium) electrodes. Eight electrode pairs were arranged at a fixed distance of 10 mm from each other within a glass vial containing the solution of proteinoid samples. The electrodes connected to an ADC-24 PicoLog data logger (Pico Technology, St. Neots, Cambridgeshire, UK). It ran in differential mode. This setup improved signal quality and reduced common-mode noise. Data acquisition happened at a rate of 1 sample per second. This rate was optimal for capturing the slow potential changes in these proteinoid systems. It also allowed for longer recording times. The voltage measurements were referenced to a common ground electrode positioned at the center of the vial. We rehydrated lyophilized proteinoid microspheres in water for the microsphere recordings. This made a uniform suspension. Fibrillar structures and mixed microsphere–fiber assemblies were made in the same medium. This setup allowed for a direct comparison of electrical properties between the different shapes. All measurements happened at a controlled temperature of 25±1 °C. They lasted from 50 to 100 h. This helped us to see both transient and steady-state electrical behaviours.

The electrical recording setup used eight pairs of Pt/Ir electrodes. They were placed at regular intervals in a glass vial with the proteinoid samples (see Figure 1). This setup allowed for simultaneous measurements on multiple channels. It captured the complex electrical behaviour of different proteinoids samples. The ADC-24 PicoLog system (Figure 1) has great recording ability. This feature was key for tracking changes in electrical potentials. These changes often displayed different phases over time periods longer than $10^{5}$ s. We took sample measurements in controlled temperatures. This helped remove thermal fluctuations as confounding factors. Therefore, the electrical patterns we observed were due to the proteinoid systems and not environmental changes.

Our study includes long observations of electrical behavior lasting up to 48 h, or 180,000 s. These long recordings reveal clear and lasting electrical patterns. Each pattern is unique to its specific shape. We see that studies on structural stability, such as time-series SEM or periodic FT-IR analysis, can provide more insights. In future work, we will use periodic sampling for morphology and spectroscopy. This will help us to connect structural changes to shifts in electrical behavior during long recordings.

We checked if the electrical phenomena came from GFD proteinoid structures, not from the measurement system. Therefore, we took similar recordings with non-polarized polymers like starch, all at the same concentrations (Figure 2). None of these controls showed the spikes or stable patterns found in GFD assemblies in water. This confirms that the electrical behaviors are intrinsic to the proteinoid structures. The starch medium changed the electrical behavior compared to pure GFD microsphere preparations. Instead of sharp spikes, it showed gradual potential changes. This control shows that the rapid spiking we saw in our main experiments comes from the GFD microsphere architectures. In our previous work, we have already established that pure water produces no significant electrical activity beyond random noise [40].

The impedance spectra were collected under controlled conditions. We used top-notch electrodes to ensure accurate and repeatable measurements of the electrochemical properties of Glu-Phe-Asp proteinoids. We used two disposable subdermal needle Pt/Ir electrodes from Technomed Europe, Kerkrade, The Netherlands. We also used one Medium Serpentine Electrode Chip, product number MS100-DIE-1EA. We chose these electrodes because they are very durable and reliable. This durability helped to keep electrical contact steady and reduce noise during impedance spectroscopy. The Pt/Ir needle electrodes allowed for accurate subdermal penetration, ensuring stable signals. The MS100 chip’s serpentine design helped it interact evenly with the proteinoid samples. This setup enabled an in-depth look at their resistive and capacitive behaviors from 1 Hz to 1 MHz.

UV–Vis spectrophotometry spectra were collected using a Jenway 7200 UV–Vis spectrophotometer, and Raman measurements were performed using confocal Raman spectroscopy (Horiba France SAS, Palaiseau, France). Impedance measurements were carried out using a PalmSens4 (Alvatek, Romsey, Hampshire, UK), a USB- and battery-powered potentiostat, galvanostat, and optional frequency response analyser (FRA) designed for electrochemical impedance spectroscopy (EIS).

## 3. Results

### 3.1. Raman Spectra Analysis of Glu-Phe-Asp Proteinoids

Raman spectroscopy is a strong tool for studying proteinoids. It helps us to understand their molecular structure and interactions. This method gives detailed insights into their vibrational modes and shapes, as shown by Zhu et al. [41]. Zhu et al. analyzed the Raman spectra of 18 amino acids and their solutions. The spectra of solids are usually sharper and more complex than those in solution. This difference comes from how molecules interact in crystalline structures [41]. This principle shows in the Raman spectra of Glu-Phe-Asp proteinoids in Figure 3. It includes spectra for microspheres, fibers, and their composite. The microspheres’ spectrum (Figure 3a) shows wider and stronger peaks. This is especially true at 1250 cm^−1^ (amide III) and 1650 cm^−1^ (amide I). These findings suggest a greater level of molecular disorder. This matches Zhu et al.’s [41] finding. They noted that solution-phase spectra usually have broader bands. This happens because of random intermolecular forces and strong motions in liquids [41]. The wider peaks in microspheres might show a more disordered structure. This could be because their spherical shape increases surface interactions and variability, much like how amino acids, such as aspartic acid and glutamic acid, act in solution. Their low solubility causes weak Raman signals and fluorescence interference [41].

The fibers’ spectrum (Figure 3b) shows sharper peaks at the same vibrational modes (1250 cm^−1^ and 1650 cm^−1^). This suggests a more ordered molecular structure, much like the crystalline amino acids studied by Zhu et al., such as glycine and alanine. They show sharp, intense bands in the solid phase, such as glycine at 894 cm^−1^ and 1327 cm^−1^ [41]. The sharp peaks of the fibers show that the proteinoid chains are well aligned. This structure might be like the order found in aliphatic amino acids, such as leucine, according to Zhu et al., who noted distinct ${\mathrm{CO}}_{2}^{-}$ deformations and C–C stretching vibrations [41]. The fibers and microspheres’ composite (Figure 3c) shows a unique spectral profile. It mixes broader peaks from microspheres with sharper features from fibers. This combination reflects a balance of molecular disorder and order. Raman spectra can track changes in shape and interactions between molecules [41]. The composite’s blended properties may make it great for biomaterial uses. For example, in tissue engineering scaffolds, a mix of flexibility from microspheres and structural strength from fibers is needed. The amide III and amide I bands in all spectra give insights into the proteinoid backbone. Their positions and intensities suggest hydrogen bonding and secondary structure, consistent with the findings of Zhu et al., who applied this analysis to collagen and identified similar amide bands at 1273 cm^−1^ and 1668 cm^−1^ [41]. Raman spectroscopy shows great promise in designing proteinoid-based materials. It reveals structural changes that affect their functions. Table 2 shows the key amide band positions for all three morphologies. The fundamental peptide bond vibrations happen at similar wavenumbers, but their intensities and bandwidths differ a lot.

### 3.2. Signal Propagation and Spiking Dynamics in Morphologically Diverse Glu-Phe-Asp Peptide Networks

The electrophysiological behaviour of Glu-Phe-Asp (GFD) peptide assemblies varies greatly. This diversity links directly to their shape and structure. This section looks at how the structure at micro- and nanoscale levels affects signal generation, propagation, and integration in various GFD architectures. Microsphere networks show quick spiking events like action potentials. They have clear depolarization–repolarization patterns and refractory periods. Fibrillar structures allow for slower potential changes and more stable behaviour over time. Mixed-morphology systems are especially interesting. Here, the interface between microspheres and fibers creates unique areas for signal processing. We analyze waveforms, spatial patterns, and voltage changes over time. This helps us find clear links between structure and electrical signals. Even simple tripeptide assemblies can create complex signaling, like advanced biological systems. These findings show key links between shape and electrical function. This can be important for early life development. It also helps in making biomimetic materials for computers.

#### 3.2.1. Spike Dynamics at the Interface of Fibrillar and Spherical Morphologies in Heterogeneous Proteinoid Assemblies

Figure 4 shows clear stabilization patterns in the extracellular potential of GFD microsphere–fiber mixtures. The recorded voltage V(t) for each channel can be characterized by a time-dependent function:(1)$$V_{i}\left(t\right)=V_{i,\infty }+\Delta V_{i}\cdot e^{-t/\tau_{i}}$$
where $V_{i,\infty }$ is the steady-state potential for channel *i*. $\Delta V_{i}$ indicates how much it differs from equilibrium at the start. $\tau_{i}$ represents the time constant for relaxation. Action potential-like spiking patterns are missing from all channels. These patterns would show rapid depolarizations, then repolarizations. Such spikes would appear as transient voltage fluctuations, following(2)$$V_{spike}\left(t\right)=V_{baseline}+A\cdot \left(e^{-\left(t-t_{0}\right)/\tau_{rise}}-e^{-\left(t-t_{0}\right)/\tau_{decay}}\right)\cdot H\left(t-t_{0}\right)$$

Here, *A* is the spike amplitude. $\tau_{rise}$ and $\tau_{decay}$ are the rise and decay time constants. $t_{0}$ marks the spike initiation time. Finally, $H(t-t_{0})$ is the Heaviside step function. The missing signatures in the GFD microsphere–fiber mix state a different electrophysiological mechanism. This contrasts with neuronal or excitable cell systems [42]. It instead reflects slow electrochemical equilibration processes in the mixed peptide assembly.

The GFD microsphere–fiber blend lacks spiking patterns. This shows key in underlying physicochemical mechanisms compared to biological neural systems. Neurons create action potentials using voltage-gated ion channels, which have specific threshold behaviours. In contrast, the GFD system relies on passive electrochemical processes. This distinction can be further analyzed across multiple dimensions:Temporal dynamics: The GFD system shows steady equilibration curves. Its time constants are in the thousands of seconds. This is very different from the millisecond-scale kinetics of neuronal action potentials. This suggests diffusion-limited processes rather than active membrane transport.Spatial heterogeneity: The steady potential differences between recording sites range from −40 mV to +45 mV. This indicates stable electrochemical microdomains. These domains have little cross-talk. This separation likely aligns with the different shapes in areas rich in microspheres and those dominated by fibers.Thermodynamic considerations: The steady shift to stable potentials means the system is near electrochemical balance. This is caused by Nernst-like potentials. This is not due to cyclic, energy-dependent actions like those in excitable membranes.Molecular interpretation: The behaviour fits a model where charged peptide assemblies create local electric fields. This occurs because of different ion adsorption and desorption effects. It does not happen through selective ion permeability, like in cellular systems.

#### 3.2.2. Intrinsic Oscillatory Behaviour in Glu-Phe-Asp Microsphere Networks

Glu-Phe-Asp (GFD) microsphere networks show spontaneous oscillations. This behaviour reveals how biomolecular structures can mimic electrical activity like neurons. Such insights could help in bio-inspired computing. Microspheres behave differently from GFD fibers. While GFD fibers show smooth potential changes, microspheres have complex spiking dynamics. This is because of their round shape and compartmentalized structure. This section looks at long-term electrophysiological recordings and detailed spike waveforms. It shows the natural oscillatory patterns in these networks.

Figure 5 shows the changes in electrical activity over time across multiple channels. In the initial phase, high-frequency spiking shows a quick buildup of ionic gradients on microsphere surfaces. The spherical shape has a high surface-to-volume ratio. This allows for efficient ion exchange between the inside and outside of the microsphere. This likely happens because GFD peptides self-assemble into porous structures. These structures create semi-permeable boundaries. These boundaries work like basic membranes. They allow certain ions to pass through and build up. This is essential for creating oscillatory behaviour. The spike amplitudes of 5–15 mV and burst patterns show that local depolarization events happen. These events are triggered by ion influx, like H^+^ or Na^+^. When they exceed a threshold, they cause quick potential changes similar to neuronal action potentials. In the transitional phase, spiking frequency drops to 0.1–0.2 Hz. This may show that ionic gradients are stabilizing or that available ions are partially depleted. Both factors lower the driving force for oscillations. In the late phase, microspheres create unique electrochemical microenvironments. The potential range is wide, from −70 to +110 mV. Different channels behave differently; for example, channel B operates at −70 mV, while channel F works at +110 mV. For example, channel F’s high potential may come from a steady flow of positive ions. Meanwhile, channel B’s negative shift could show anion buildup or proton loss. Microspheres differ from GFD fibers. GFD fibers spread ionic gradients over long distances. In contrast, microspheres keep these gradients contained. This helps maintain localized potential differences that change over time.

Figure 6 shows spike waveforms more closely. It reveals different mechanisms in the channels. Channel A shows upward spikes of 2–4 mV. These spikes and the rising frequency suggest that ionic conductivity is improving. This may be due to microsphere swelling or the buildup of surface charge over time. Channel C shows a quick rise and a slow fall (3–5 mV). This pattern is like the typical depolarization–repolarization cycles. First, ions rush in (depolarization). Then, they slowly flow out or diffuse (repolarization).

The spiking dynamics show that GFD microspheres create compartments. These compartments can support oscillatory behaviour. We can model this using a simplified Hodgkin–Huxley-like framework, where the membrane potential V(t) evolves as follows:(3)$$C_{m}\frac{dV}{dt}=-g_{ion}\left(V-V_{eq}\right)+I_{ion}\left(t\right),$$
where $C_{m}$ is the effective capacitance of the microsphere surface, $g_{ion}$ is the ionic conductance, $V_{eq}$ is the equilibrium potential, and $I_{ion}\left(t\right)$ represents time-varying ion currents. The rapid rise (1–2 ms) indicates a high $g_{ion}$, driven by the microsphere’s ability to concentrate ions at its boundary.

Channel D’s step-like transitions hint at a quantized release or uptake of ions. This may happen because of distinct pore openings or changes in the GFD matrix. Channel E’s bistable switching (8–10 mV swings) shows a nonlinear feedback. The potential moves between two stable states.(4)$$\frac{dV}{dt}=-\alpha V\left(V-V_{1}\right)\left(V-V_{2}\right)+\eta \left(t\right),$$
where $V_{1}$ and $V_{2}$ are stable potentials (e.g., −10 mV and +5 mV), α controls transition rates, and η(t) is a stochastic noise term from ionic fluctuations. Channel H shows high-frequency, low-amplitude spikes (0.6 Hz, 1–2 mV) with bursts. This suggests at a pacemaker-like rhythm. It may come from cyclic proton movement on the microsphere’s surface.

These mechanisms hinge on the microsphere’s geometry. The spherical shape maximizes curvature, creating steep electrochemical gradients over short distances. GFD fibers are different. Gradients fade along their length, stopping spike formation. The Nernst–Planck equation can approximate the ionic flux *J* across microsphere surfaces.(5)$$J=-D\frac{\partial C}{\partial r}+\frac{zF}{RT}DC\frac{\partial V}{\partial r},$$
where *D* is the diffusion coefficient, *C* is ion concentration, *r* is the radial distance, *z* is ion charge, *F* is Faraday’s constant, *R* is the gas constant, and *T* is temperature. The second term, driven by the potential gradient $\frac{\partial V}{\partial r}$, is amplified by the microsphere’s compact structure, enabling rapid ion movements that sustain oscillations.

In summary, the round shape of GFD microspheres helps with compartmentalization. This allows for swift ion exchange, leading to action potential-like events that do not happen in fibrillar structures. This structural distinction underpins their potential for complex signal processing.

#### 3.2.3. Distinctive Spiking Behaviour in Glu-Phe-Asp Fiber Networks

Glu-Phe-Asp (GFD) networks behave quite differently based on their shape. Microspheres create quick spikes that resemble action potentials. In contrast, fibers show smooth, steady potential changes and slow oscillations. These differences come from their structures. Microspheres have spherical compartments, while fibers have long, spread-out gradients. This section looks at long-term recordings and high-resolution waveforms from both systems (refer to Figure 5, Figure 6, Figure 7 and Figure 8 for more information). This helps clarify the underlying mechanisms. The differences are clear in Figure 5 and Figure 7. Microspheres start with a burst of fast spikes because their small, round shape acts like a tiny container. Ions are charged particles, like hydrogen or sodium. They can build up quickly inside or around surfaces. Then, they rush across these surfaces, causing sudden voltage jumps. Imagine water sloshing in a small cup; it moves fast and splashes easily. Over time, as the transitional phase kicks in, these bursts slow down, possibly because the ion supply steadies or is used up. In the late phase, each channel finds its own pattern. Some channels hold high voltages, while others drop low. This shows how microspheres can form distinct “worlds” of electrical activity.

Fibers behave more like a long, slow river. In Figure 7, the initial phase shows large voltage swings. They grow gradually as ions spread out along the fiber’s length. Fibers do not trap ions in one spot. Instead, the charges spread along their thread-like structure. This smooths out quick changes. In the transition phase, these voltages settle into position. By the extended equilibrium phase, they stabilize into bands that hardly change for more than 200,000 s. This steadiness means that fibers can hold electrical patterns longer than the spiky microspheres.

The close-up views in Figure 6 and Figure 8 reveal even more about what is happening. In microspheres (Figure 6), Channel C’s spikes rise sharply and then fade. They look like a spark that ignites quickly and hangs around. Ions rush in and then flow out. Channel E flips between high and low voltages, hinting at a back-and-forth tug-of-war inside the microsphere. Channel H fires off tiny, fast spikes in bursts, acting like a little pacemaker. This variety happens because microspheres can trap ions in tiny pockets. This allows the ions to build up and release quickly from their curved surfaces. The fibers tell a different story in Figure 8. Early on, Channel F creeps up slowly from 41 to 47 mV with little ripples, showing ions drifting along the fiber without sudden bursts. Later, in the middle phase, bigger waves appear, rising and falling over thousands of seconds. These waves do not spike—they are more like a lazy tide, moving ions in large, coordinated groups across the fiber’s length. This slow rhythm occurs because fibers cannot concentrate ions in one area. Their stretched shape spreads ions out, which smooths the electrical flow. The root of these differences lies in how microspheres and fibers handle ions. Microspheres, being small and round, act like tiny batteries; ions pile up fast, then discharge in sharp bursts, creating spikes. Fibers, being long and thin, work more like a steady conveyor belt; ions move along slowly and evenly, leading to gradual shifts or gentle waves. Microspheres are great for quick, changing signals. Fibers are best for steady, long-lasting patterns. Together, these traits show how GFD networks can adapt their electrical behaviour simply by changing shape.

Mixing microspheres and fibers, as shown in Figure 4, finds a balance. Early on, the voltages wiggle and shift as the two shapes jostle for balance. After 10,000 s, things calm down, spreading across a range from −40 mV to +45 mV. Some channels, like E, lock into a high voltage, likely from microsphere patches pulling in ions fast. Others, like H, dip low, where fibers dominate and spread ions out. This mix does not spike like pure microspheres or stay still like pure fibers. Instead, it is a blend in which each part affects the others. This creates a network where quick bursts and slow drifts work together in a steady, varied vibration. Microspheres quickly grab ions, spiking when possible. Fibers stretch ions into stable, lasting patterns. They create a system with different zones. Some channels are alive with microsphere energy, while others are calm and fiber-like. This balance lasts over 60,000 s. It shows that the mixture can handle both changing signals and stable memory. This is different from the simple behaviours of pure microspheres or fibers. The system shows a complex frequency spectrum. It has both high-frequency oscillations and low-frequency baseline shifts. Signal features are connected to the shape and surface properties of spheres and fibers.

### 3.3. Morphology of Glu:Phe:Asp Proteinoids

Glu-Phe-Asp (GFD) fibrillar networks have a complex structure. This structure leads to their unique electrical behaviour. This includes smooth voltage shifts and slow, wave-like oscillations. You can see these in Figure 7 and Figure 8. SEM images in Figure 9 show how these fibers arrange. They range from tight threads to tangled webs. This structure—fibers, strands, and meshes—affects how ions move and voltages settle. It is unlike the quick spikes seen in microspheres (Figure 5). We connect these physical forms to their electrical signatures. This shows how fiber design affects stability and speed.

#### 3.3.1. Hierarchical Organization and Structural Characteristics of Self-Assembled Glu-Phe-Asp Fibrillar Networks

High-resolution electron microscopy showed that Glu-Phe-Asp (GFD) tripeptides can form fibrillar structures. These structures have a unique hierarchical organization. As shown in Figure 9a, individual fibers exhibit well-defined dimensions with lengths of 1.702μm and widths of 0.159μm, resulting in aspect ratios exceeding 10. These fibers show radial patterns. This suggests they form from central points. This property might affect their electrochemical behaviour. The fibrillar structures show a wide range of lengths. In Figure 9b, some fibers stretch to 23.162μm. They also have narrow widths of about 0.650μm. This dimensional plasticity allows for large networks that can cover great distances. This might help explain the long-range electrical conductivity seen in GFD fiber systems. In this study, these structural features stayed the same in both standard and high-vacuum conditions. This shows they are intrinsic, not just effects of the environment. SEM analysis (Figure 9c) revealed that the smallest fibrillar elements are nanoscale in size. Their minimum widths are about 80 nm. The lamellar structure seen here shows how GFD peptides are organized in the fibers. This ordered arrangement likely helps give them their special electrical properties. The consistent width-to-length ratios seen at different scales suggest a key self-assembly rule that controls GFD fiber formation. Connecting individual fibers into complex networks, shown in Figure 9d, creates mesh-like structures. These structures have clear branching points. This pattern forms a steady fibrillar matrix.

The SEM images in Figure 9 peel back layers of the GFD fibrillar world. Figure 9a shows individual fibers that measure approximately 1.7μm in length and 0.16μm in width, tightly packed together like rods in a cluster. Their radial spread from central points suggests they grow outward from a starting seed, creating a dense mat. This compactness likely slows ion movement. It spreads charges evenly along their length (imagine water seeping through a thick sponge instead of splashing out). This relates to the slow voltage rise in Figure 8 (subfigure a). Channel F increases from 41 to 47 mV over 20,000 s. The change shows small shifts, not sharp spikes. The steady, banded potentials in Figure 7 stay between +40 and +65 mV or −60 and −90 mV for hours. This shows even distribution and keeps ions in stable patterns.

Figure 9b zooms out to reveal longer fibers, stretching over 23μm with a width of 0.65μm, growing in clear, straight lines. Their high aspect ratio—about 35 times longer than wide—means ions have a long path to travel, smoothing out any sudden shocks. This matches the slow, repeating waves shown in Figure 8b. Here, voltages rise and fall by 3–5 mV over thousands of seconds, like a tide coming in and out. Figure 9c dives deeper, showing ultrafine fibers down to 80 nm wide, with layered, plate-like details inside. These tiny strands can serve as narrow channels. They guide ions in thin streams. This helps reinforce the smooth flow in the fiber network. Figure 9d shows a tangled mesh. This hints at a cooperative system where fibers connect to share electrical loads. This helps support long-term stability in Figure 7.

This structural variety explains why fibers lack the microsphere’s quick sparks (Figure 6). Microspheres trap ions in tight, round pockets for quick bursts. In contrast, fibers stretch the ions out. This allows charges to drift slowly along their length or weave through the fibers. To quantify this, the fiber aspect ratio influences ion diffusion time—longer fibers mean slower shifts. For a fiber of length *L* and width *W*, the diffusion timescale stretches as follows:(6)$$\begin{array}{c} \tau_{d}\propto \frac{L^{2}}{D}, \end{array}$$
where *D* is how fast ions move. In Figure 9b, a 23μm fiber takes far longer to shift voltage than a 1.7μm one in (a), aligning with the drawn-out waves in Figure 8. The network’s mesh in (d) might also dampen rapid changes. This mesh boosts the system’s ability to share electrical load across many fibers, smoothing out jolts. We can think of its conductance—the ease of current flow—as growing with the number of connections, roughly captured by(7)$$\begin{array}{c} G\propto \frac{N}{L}, \end{array}$$
where *N* is the number of fiber intersections and *L* their average length. More connections in subfigure (d)’s tangle mean steadier currents, matching the hours-long stable bands in Figure 7, much like +40 to +65 mV holding firm. Unlike microspheres’ compact chaos, fibers offer a calm, enduring canvas for electrical memory.

**Figure 9 biomimetics-10-00360-f009:**
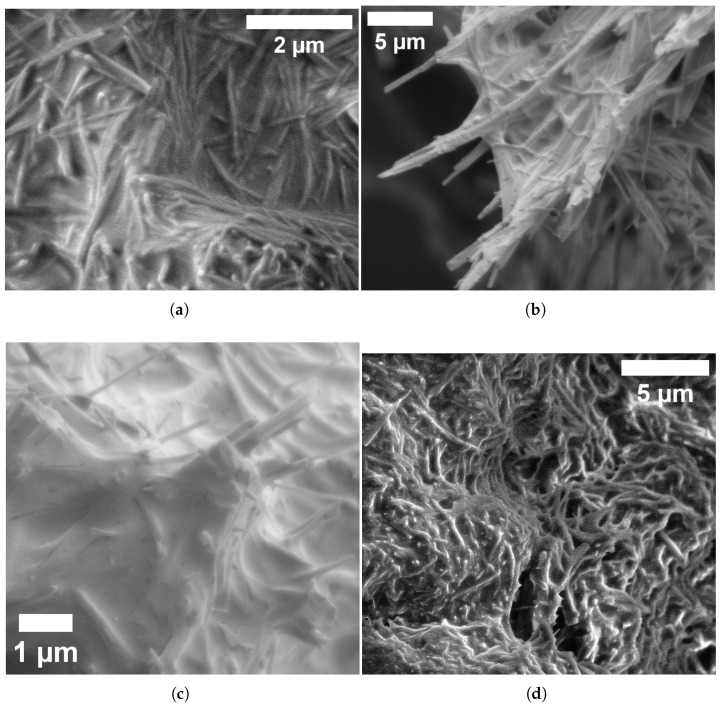
SEM images show the structure of Glu-Phe-Asp fibrillar networks. They reveal different levels of detail based on magnification and sample preparation. (**a**) High-resolution images show individual fibers. Each fiber is about 1.702μm long and 0.159μm wide. They have a typical aspect ratio of around 10.7 and are tightly packed. The radial distribution of fibers shows that nucleation happens from central points. The horizontal field width (HFW) is 8.29μm. This was measured using an LFD detector at HV = 2.00 kV, with a magnification of 25,000× and a chamber pressure of $9.82\times 10^{-1}$ Torr. (**b**) Extended fiber structures show pronounced elongation (length = 23.162μm, width = 0.650μm) and directional growth patterns. The high aspect ratio (about 35.6) shows key structural properties. This is true even under high-vacuum conditions. The ETD detector operates at a pressure of $1.27\times 10^{-5}$ Torr. With an HFW of 33.4μm and a magnification of 6209×, these properties remain unaffected by the environment. (**c**) Ultrafine fibrillar elements (length = 1.162μm, width = 0.080μm) exhibiting nanoscale dimensions with minimum width approaching 80 nm. The high-resolution imaging shows lamellar substructure in each fiber. It uses an ETD detector with 20,000× magnification and an HFW of 10.4μm and operates at a pressure of $2.49\times 10^{-5}$ Torr. (**d**) Complex network design shows a tangled fibrillar structure. It has clear branching points and fiber sizes that vary. It operates at HV = 5.00 kV, with a magnification of 6289×. Its HFW is 39.0μm, and the pressure is $9.82\times 10^{-1}$ Torr. This setup demonstrates how single fibers collaborate. This may affect the overall electrical properties of the network.

#### 3.3.2. Morphological Interfaces in Hybrid Glu-Phe-Asp Systems: Cooperative Assembly Dynamics Between Fibrillar and Spherical Architectures

The assembly behaviour of Glu-Phe-Asp (GFD) proteinoids was studied using high-resolution electron microscopy. This revealed complex interactions between microspheres and fibrillar forms. Perforated microspheres (d=6.576μm) have large central holes ($d_{hole}=3.047\;\mu$m). They connect directly with fibrillar extensions that are 5.930μm long, as shown in Figure 10a. These contact regions are possible electrochemical junction points. Here, the different electrical properties of each morphology may interact. This could explain the complex spiking patterns seen in mixed systems. Higher-magnification imaging (Figure 10b) showed clear surface interactions at the edges of various shapes. There was a noticeable difference in electron density between microspheres and fibers. This suggests that the molecular organization varies. This interface region helps shift from the quick, spike-producing traits of microspheres to the slower changes seen in fibrillar networks. The microsphere population in mixed groups showed great size variety and a clear structure (Figure 10c). Multi-generational structures have diameters from 2.329μm to 25.373μm. This range shows different development stages. Smaller structures may indicate early formation processes. This size distribution leads to different compartment volumes. These volumes can support varying electrochemical dynamics in the same network. The discovery of advanced encapsulation phenomena was fascinating (see Figure 10d). Large hollow microspheres (d=33.152μm) held smaller spherical structures inside them (d=1.5–4.5μm). This nested design shows a rare self-organization in synthetic systems. It may also allow for separate spaces for different electrochemical processes. Hierarchical compartmentalization may boost the complex signal processing in mixed GFD networks. It does this by forming electrically isolated areas that have unique response traits. Both spherical and fibrillar elements are present in various samples. This shows that GFD proteinoids naturally assemble in different forms. This is not just a result of preparation artifacts. This structural diversity links to the complex electrical behaviours we recorded. In regions with many microspheres, we see unique spiking patterns. In contrast, areas rich in fibers show gradual potential changes.

The SEM images show how microspheres and fibers interact. This highlights a lively assembly process. In Figure 10a, the big hole in the microsphere acts like a gateway. Fibrillar extensions grow outward from it. This suggests that fibers come from spherical hubs. This connection acts like a bridge. It may move ions from the fast-spiking microsphere areas to the slower fiber regions. This improves the mixed system’s ability to switch between different signal types. Figure 10b shows the rough edges where microspheres touch fibers. Darker and lighter patches suggest different material densities. This might mean denser peptide packing in fibers and looser arrangements in microspheres. This boundary can act as a buffer zone. It helps transition from the microsphere’s quick bursts to the fiber’s slow drifts. This adds flexibility to the network’s electrical response.

Figure 10c shows a lively microsphere community. Sizes range from small buds to larger spheres, suggesting a living growth cycle. The small structures starting to form may signal early activity. The larger ones store more charge. This creates a gradient of electrochemical activity in the network. This size gradient likely affects the different voltage zones in mixed systems. Some areas buzz with microsphere energy, while others are calm and fiber-like. In Figure 10d, the nested microspheres inside a giant hollow shell resemble a Russian doll, offering a layered sanctuary for ions. This encapsulation can separate various electrochemical processes. Inner spheres spark for a short moment, while outer shells remain steady. This boosts the network’s ability to process complex signals. These morphological features show that GFD hybrids are adaptable. Shape-driven interactions create a mix of electrical behaviours, ranging from quick pulses to lasting patterns.

Figure 11 shows key SEM evidence of L-ASP’s ability to self-assemble in different forms at various scales. Panel (a) shows a round supramolecular structure with a clear hierarchy at a 20 μm scale. It has cauliflower-like growth patterns, which suggest that nucleation drives the assembly process. The rough, clustered surface has a varied texture. It shows differences in electron density, hinting at different peptide packing densities. Panel (b) shows a close-up view of a complex network of tiny pores. These pores are about 50–200 nm wide. They likely help with ion exchange, which is important for the electrochemical behavior we see. These nanoscale cavities boost the surface-to-volume ratio. This improvement boosts capacitive properties. It also explains the measured impedance values of 6646.282 ± 178.664 Ohm in microsphere preparations.

Panel (c) shows a larger field of view at 50 μm. This reveals a complex 3D structure. The interconnected fibrillar elements create a mesh-like superstructure. This layout creates many junction points. These points probably act as electrochemical interfaces. They are key for the action potential-like spiking behavior we observe. The high-magnification image in panel (d) shows a clear spheroidal structure. It has a smooth surface and is surrounded by smaller particles. The clear boundary shown here helps to create compartments. These compartments generate steep ion gradients. This is important for quick depolarization events, as shown in Figure 6. These structural elements, such as nanoporous surfaces and special compartments, form the basis for electrical changes. This is what sets apart the microsphere, fibrillar, and mixed morphologies we observed in our experiments.

### 3.4. Electrochemical Dynamics of GFD Proteinoids: Time-Dependent Impedance, Capacitance, and Resistance Analysis

We studied the electrical behaviors of Glu-Phe-Asp (GFD) proteinoids. We studied the electrochemical properties of fibers, microspheres, and their mixture for 1000 s. This analysis was carried out with samples taken every second. Using impedance (Z/Ohm), capacitance (Cs/μF), and resistance (Idc/mΩ) data, we quantified the dynamic responses of each morphology. This analysis helps us to understand spiking patterns in microspheres (Figure 6), gradual waves in fibers (Figure 8), and hybrid stability in the mixture (Figure 4). It shows how structural polymorphism affects charge storage, ion flow, and resistance over time.

We studied the electrical behaviours of Glu-Phe-Asp (GFD) proteinoids. We looked at the electrochemical properties of fibers, microspheres, and their mix. This analysis lasted 1000 s, with samples taken every second. Using impedance (Z/Ohm), capacitance (Cs/μF), and resistance (Idc/mΩ) data, we quantified the dynamic responses of each morphology. Figure 12 shows the average electrical properties. These properties create clear profiles. They match the spiking patterns in microspheres (Figure 6). They also match the gradual waves in fibers (Figure 8) and the hybrid stability in the mixture (Figure 4). The statistics in Table 3 show capacitance values from code analysis. These values match those in Figure 12. Fibers have a high capacitance of 9.912±0.171
μF. In contrast, microspheres have a much lower capacitance of 1.926±5.735
μF. The mixture’s capacitance is 3.328±0.076
μF, which is between the two. A plot of capacitance over time backs up these findings. It shows that fibers keep a stable, high capacitance, which means they can store charge effectively across their structure. Microspheres show quick spikes, hitting 162.514 μF, as seen in Figure 12. This reflects fast charge buildup events linked to their spiking behaviour (Figure 6). The mixture shows an intermediate capacitance, balancing the two morphologies. Impedance profiles further highlight morphological differences. Figure 12 shows that microspheres have a much higher impedance of 6646.282±178.664 Ohm. In contrast, fibers measure 209.400±0.286 Ohm, and the mixture has 404.235±1.091 Ohm. The impedance plot supports this. It shows that microspheres have a higher impedance that changes, likely because of quick ion movements during spikes. Fibers have lower impedance, which helps ions flow smoothly. This supports the gradual voltage waves seen in Figure 8. The mixture’s intermediate impedance shows its hybrid nature. This is clear in the stable-voltage zones (Figure 4). Resistance measurements, scaled to mΩ, reveal an inverse relationship with capacitance. Figure 12 shows that fibers have the highest resistance at 163,067.613±9253.064 mΩ. Next is the mixture at 60,424.487±1293.986 mΩ. Microspheres have the lowest resistance at 15,830.739±652.514 mΩ. The resistance plot shows this trend too. Fibers have a steady, high resistance. This limits current flow and matches their ion dynamics. Microspheres have the lowest resistance. This allows ions to move more easily. As a result, we see the rapid spikes shown in Figure 6. The mixture’s intermediate resistance highlights its synergistic properties. It balances both dynamic and stable behaviours. These electrochemical signatures show how GFD proteinoid shape affects electrical behaviour. We can see them in Figure 12. Fibrillar structures are great for storing charge, but they resist current flow. Microspheres allow for quick ionic transport, even with high impedance. Mixed systems show how these features work together, which can inspire new bio-based applications.

Figure 13 displays the Bode plot analysis of different GFD proteinoid morphologies. Each morphology exhibits unique electrical characteristics that vary with frequency, supporting our proposed structure–function relationships. The impedance magnitude (|Z|) and phase response curves show clear morphological differences across the frequency range of $10^{-2}$ to $10^{6}$ Hz. Fibrillar structures exhibit the lowest impedance, dropping to approximately 100Ω at high frequencies. They also show minimal phase shift, indicating predominantly resistive behavior. Their elongated shape facilitates charge movement along well-defined paths. Microspheres demonstrate significantly higher impedance, approximately $10^{4}\;\Omega$ across most of the frequency spectrum. They also present a strong negative phase shift, approaching $-50^{\circ }$ at $10^{6}$ Hz. This behavior suggests the presence of capacitive elements, likely due to their spherical architecture and tendency to accumulate surface charge. The fiber–microsphere composite exhibits an intermediate impedance profile and a complex phase response that includes both resistive and capacitive components. This observation reinforces our hypothesis that mixed morphologies yield electrochemical systems with hybrid electrical properties. The frequency-domain characteristics highlight that the electrophysiological signatures—fast spiking in microspheres and slow oscillatory waves in fibrils—stem from their intrinsic impedance behaviors rather than from measurement artifacts or environmental interference.

Nyquist plots in Figure 14 show important details about the electrochemical behavior of Glu-Phe-Asp proteinoids in various forms. For fibers, the compressed semicircle shows that $Z^{\prime }$ ranges from 0.180 to 0.181 kΩ. Meanwhile, $-Z^{\prime \prime }$ varies from −0.015 to −0.008 kΩ. This indicates a mainly resistive response with low charge transfer resistance. So, a simple resistor model likely explains the impedance behavior. This can be represented by a basic circuit with one resistor. In Equation (Equation 8), the impedance *Z* is real and does not depend on frequency. The fibers and microspheres composite shows $Z^{\prime }$ ranging from 0.418 to 0.455 kΩ and $-Z^{\prime \prime }$ from −0.071 to −0.061 kΩ. This results in a clearer semicircle, which means a balanced resistive–capacitive response. This behavior hints at a series circuit with a resistor and a capacitor. We can explore this by zooming into the area $Z^{\prime }$: 0.400 to 0.450 kΩ and $-Z^{\prime \prime }$: −0.080 to −0.040 kΩ.(8)Z=R

The microspheres plot in Figure 14 shows a larger semicircle. Here, $Z^{\prime }$ ranges from 3.162 to 5.024 kΩ, and $-Z^{\prime \prime }$ goes from −4.492 to −4.064 kΩ. This indicates a significant capacitive impedance, aligning with a parallel RC circuit. This behavior emerges when we zoom into the region $Z^{\prime }$: 3.0 to 3.5 kΩ, and $-Z^{\prime \prime }$: −4.5 to −3.5 kΩ. The semicircle’s shape reveals the capacitive effect, likely from surface-bound ionic interactions. The impedance of a parallel RC circuit is shown in Equation (Equation 9). Here, *R* stands for resistance, *C* for capacitance, and ω the angular frequency. These distinct electrochemical responses—resistive for fibers, hybrid for fibers and microspheres, and capacitive for microspheres—suggest tailored applications in bioelectronics, such as conductive pathways, hybrid interfaces, or capacitive sensors, respectively, which can be further analyzed using the interactive features of the plots.(9)$$Z=\frac{R}{1+j\omega RC}$$

### 3.5. UV–Visible Spectroscopic Fingerprints of Polymorphic GFD Proteinoid Assemblies

UV–Visible spectroscopy gives key insights into the electronic structure and organization of GFD proteinoid assemblies. It complements our studies on morphology and electrophysiology. Figure 15 shows that the extended spectral range (350–1050 nm) reveals clear differences in optical behaviors among the three morphological variants. The first absorption edge seen at 350–375 nm in all samples is due to π–$\pi^{*}$ transitions in the aromatic phenylalanine residues. This acts as a common molecular signature. However, aside from this shared feature, each morphology differs greatly. This shows their unique supramolecular organization. The mixture shows a sudden change in light behavior at about 375 nm. It shifts from strong absorption to full transparency in the visible and near-infrared areas. This sharp change points to a unique electronic structure at the interface between fibers and microspheres. Here, uneven charge distributions might lead to optical band gaps that are not found in uniform structures.

The close-up view of the 380–850 nm range in Figure 16 shows the electronic properties of microspheres. This difference stands out when compared to other shapes. The spectral profile of microspheres shows several absorption bands. There is a sharp peak at 410 nm. This peak likely relates to $n-\pi^{*}$ transitions in the carbonyl groups of glutamic and aspartic acid residues. The main absorption peaks are around 570–630 nm. These peaks come from electronic transitions in the peptide backbone. There is also a smaller peak near 750 nm, which might suggest extended conjugation networks. These spectral features match our model. Microspheres are compartmentalized structures. They have mixed charge distributions. This setup helps them support quick, spike-like electrical discharges. The complex absorption profile shows that there are many electronic energy levels. These levels may explain the threshold-dependent behaviors seen in our electrophysiological recordings.

Unlike the rich features of microspheres, the fiber samples in Figure 16 show little absorption beyond the edge. This suggests a more uniform electronic structure across their long design. This simple pattern connects to our finding of gradual, wave-like electrical behavior in fibrillar assemblies. Here, charge spreads evenly over larger distances. The lack of distinct absorption bands in the visible range means the fibers do not have the specific electronic states needed for quick charge buildup and release. Their electronic structure seems to allow continuous charge flow along the fiber axis. This explains the steady potential changes we observed in our long-term recordings. This behavior, which depends on structure, supports our model of morphology-driven electrical polymorphism.

The 50:50 mixture shows fascinating spectral behavior in both Figure 15 and Figure 16. This behavior cannot be explained as just a simple mix of the pure morphologies. The sudden shift from absorption to transparency hints at new optical properties. These come from the complex interactions between microspheres and fibers. These may involve constructive or destructive interference at structural interfaces. They can also include quantum confinement effects at junctions. Additionally, charge transfer complexes can form between different morphologies. The mixture shows optical behavior that matches its electrical properties. Here, spike-like events and gradual potential changes exist together in a balanced system. This evidence supports our main idea. Structural changes in GFD proteinoids lead to varied functions. This happens through different electronic structures based on their shape. So, these self-assembling systems show great potential for biomimetic signal processing uses.

## 4. Discussion

Glu-Phe-Asp (GFD) proteinoid systems show various shapes and electrical behaviours. These differences show how structure relates to function in self-assembling biomimetic materials. Our findings show that GFD peptides can take on different stable shapes, like microspheres, fibers, and mixed forms. Each shape has its own unique electrophysiological signature that relates to its structure.

### 4.1. Structure–Function Relationships in GFD Assemblies

Microspheres and fibers have very different electrical properties. This shows they likely have unique ways of handling ions and transporting them. Microspheres have clear boundaries and internal volumes. They create quick, spike-like potentials that are like action potentials in biological neurons. This behaviour likely comes from their round shape. This shape creates small enclosed areas that help form local ion gradients. These gradients make quick discharge events easier. The size distribution in microsphere populations (Figure 10c) can add to functional diversity. Different-sized compartments may show different threshold potentials and refractory periods.

Fibrillar structures show gradual and steady potential changes. They do not have distinct spiking events (Figure 7). This behaviour matches their extended shape (Figure 9). It allows ions to spread uniformly along the fiber lengths instead of piling up in segments. Fibrillar networks lack clear boundaries. This makes it hard to create the steep transmembrane gradients needed for spike generation. These structures are great at keeping stable potential distributions for long times. They may act as “memory” elements in mixed networks.

Microspheres and fibers have different electrical properties. They resemble the specialized functions found in biological neural parts in a detailed manner. Microspheres can generate spikes like neuronal cell bodies. In contrast, fibers spread signals gradually, like axonal or dendritic processes. This new biomimicry is impressive. These structures form themselves using simple tripeptide blocks. They do this without any specific design or outside input.

Glu-Phe-Asp (GFD) proteinoids show different electrophysiological behaviours. In microspheres, they create rapid spikes. In fibers, they produce gradual waves. When mixed, they form hybrid patterns. This shows how structure affects function. Microspheres are tight and round. They create quick voltage spikes. Figure 6 shows this, especially in channels like C, which have sharp jumps of 3–5 mV. Fibers stretch and weave together, creating smooth voltage shifts. In Figure 8, you can see Channel F drifting by 6 mV over 20,000 s, showing gentle waves. The 50:50 mixture balances these traits. It settles into a steady range of −40 to +45 mV across different zones (Figure 4). This mix combines the microsphere’s vibe with the fiber’s calm. Figure 17 illustrates why these patterns emerge. In microspheres, the process happens in stages. First, ions balance out. Then, they gather in tight spaces. Finally, they burst past a threshold, causing rapid spikes. Fibers spread ions along their length. This allows charges to drift at a gradual pace. This schematic shows key differences. Microspheres use compartmentalized bursts based on thresholds. Fibers spread ions for a gradual change. These insights show how the mixed system works. In some areas, there are many microspheres, which cause spikes. In other areas, fibers are more common, leading to stability. This creates a network that balances quick signals with lasting patterns, as seen in our recordings.

SEM images show the different shapes. Our findings reveal the molecular interactions that cause the structural changes in GFD proteinoids. The presence of microsphere or fibrillar structures likely comes from a careful balance of non-covalent forces. We believe that in microspheres, hydrophobic interactions pull phenylalanine residues together to form a spherical core. Meanwhile, the charged glutamic and aspartic acid residues face the water. This setup creates the electrochemical gradients needed for spiking behavior. This setup maximizes charge separation over a short distance. This explains the quick potential changes we saw in our measurements. The high impedance (6646.282 ± 178.664 Ohm) and low capacitance (1.926 ± 5.735 μF) values support a model of compact, compartmentalized charge distribution. However, fibrillar assembly seems to follow different molecular driving forces. We believe that directional hydrogen bonds between peptide backbones and π–π stacking between nearby phenylalanine residues help promote linear growth. This prevents spherical clumping. This architecture spreads charged residues along the fiber. It creates the gradual electrochemical gradients seen in our recordings. The significantly higher capacitance (9.912 ± 0.171 μF) and lower impedance (209.400 ± 0.286 Ohm) of fibrillar structures support this model of extended charge storage capability. Future work using methods like circular dichroism spectroscopy or molecular dynamics simulations can improve our understanding of these molecular interactions. This may help to better control morphology. We can achieve this by changing specific amino acids or adjusting environmental factors.

The GFD microspheres quickly create short voltage spikes. These spikes have clear phases of depolarization and repolarization. They also show specific amplitude thresholds and refractory periods that look like neuronal signals. These events probably happen in different ways. They involve non-specific ion adsorption and desorption at charged surfaces. Different diffusion rates occur across semi-permeable barriers formed by condensed polypeptide networks. Lastly, charge can accumulate in small, compartmentalized areas. The electrochemical data in Figure 10 and Table 2 back this alternative mechanism. They show that the microspheres have high impedance (6646.282 ± 178.664 Ohm) and low capacitance (1.926 ± 5.735 μF). This combination allows for quick charge redistributions without needing special channel proteins.

Spike generation in our GFD proteinoid systems is probabilistic. This is a key difference from the deterministic action potentials found in biological cells. Biological neurons rely on coordinated, threshold-based voltage-gated ion channels for reliable spike propagation. In contrast, our proteinoid microspheres produce random discharge events, which likely result from fluctuations in local ion concentrations across their semi-permeable boundaries. We identified several factors that influence spike probability without requiring structural changes, despite the probabilistic nature of the system.

Ionic strength of the medium: higher ion concentrations increase spike frequency.Temperature changes: warmer temperatures (25 °C to 37 °C) elevate discharge rates by enhancing molecular motion and ion diffusion.pH levels: acidic environments lead to more frequent and larger spikes in microspheres, likely due to changes in protonation states of glutamic and aspartic acid residues.Presence of divalent cations: ions such as ${\mathrm{Ca}}^{2+}$ and ${\mathrm{Mg}}^{2+}$ significantly increase spike probability by forming charge bridges that facilitate coordinated ion movement.

These modulatory factors are reversible and can be applied transiently, demonstrating that these simple systems are capable of dynamic, environmentally responsive behavior.

These modulation mechanisms show interesting similarities to the taxis behaviors of early life forms. They also raise important questions about evolution. Our proteinoid systems respond to environmental factors. This response could explain chemotaxis-like behaviors in protocellular structures. These structures may have come before more complex biological signaling systems. Particularly fascinating is the potential connection to radiotaxis. While we have not specifically tested the effects of ionizing radiation on our GFD systems, the work by Rowe et al. [43] and Atri et al. [44] on radiation responses in simple biological systems offers a provocative direction for future investigation. Ionizing radiation creates free radicals and reactive oxygen species. These can change charge distributions locally. Therefore, it is likely that our proteinoid microspheres show different spiking patterns when exposed to radiation. This might be like how extremophiles like Desulforudis audaxviator work. It seems to use radiolysis products in its metabolism. If our proteinoid systems respond to radiation-induced changes, it suggests a deeper link. This connection could exist between simple self-assembled peptide structures and early environmental sensing. These mechanisms may have operated in prebiotic or early biotic environments.

### 4.2. Interface Dynamics in Heterogeneous Assemblies

The interfaces between microspheres and fibers (Figure 10a,b) are intriguing areas. Here, the unique electrical properties of each sample come together. These junction regions probably help with complex signal processing. They do this by matching impedance, filtering signals, or creating modulation effects. The presence of perforated microspheres linked to fibrillar extensions hints at special signal paths between different areas.

In mixed systems, we saw a range of electrochemical behaviours. These behaviours cannot be explained just by adding the properties of microspheres and fibers together. Instead, the 50:50 mixture exhibits emergent dynamics suggesting cooperative interactions between components. Mixed networks can combine quick spikes with steady potential changes. This suggests they have basic information-processing capacities that are better than each type of structure alone.

Some advanced microsphere assemblies show a nested architecture (Figure 10d). This adds complexity with hierarchical compartmentalization. These structures have microspheres inside larger hollow spheres. This design can create isolated microenvironments with unique electrochemical properties. These properties might help with better signal integration or temporal processing.

The electrical activity in GFD microspheres works like neuromuscular control systems. It depends on complex ion channel dynamics that resemble how biological signals are transmitted. Voltage-gated sodium and calcium channels cause rapid depolarization spikes in microspheres. This closely resembles action potentials at neuromuscular junctions (NMJs), as explained by Catterall et al. [45]. These channels help ions move accurately. This fast signal flow is key for biomimetic uses. Presynaptic calcium channels help release neurotransmitters. According to Dolphin et al. [46], they assist vesicles in fusing, much like how ion flows in microspheres can trigger electrical events. Furthermore, ligand-gated ion channels, as explored by Lev et al. [47], undergo structural changes. This allows for quick signal transduction, which explains the fast electrical responses seen in both microspheres and NMJs. Anesthetics can change these channels [40,48]. This shows how ionic and molecular interactions affect signal flow. These insights help in creating responsive synthetic systems. Disruptions in channel function, seen in autoimmune channelopathies [49], show how fragile bioelectrical transmission is. This highlights the need for strong channel mimics in microsphere-based technologies. GFD microspheres show great promise for creating advanced biomimetic systems. They can help with signal processing and control. Moreover, they may have uses in neural interfaces and synthetic biology.

### 4.3. Potential Applications in Neuromorphic Computing

Proteinoid assemblies align well with neuromorphic computing due to their bio-inspired properties. Wang et al. highlight that protein-based materials provide biocompatibility, hierarchical structures, and sustainability [36]. These features make them perfect for mimicking biological neural systems. GFD proteinoids could lead to energy-efficient, brain-like computing systems. These systems may process information in parallel and adaptively. This can be helpful for tasks such as pattern recognition and sensory processing. Huh et al. also mention that materials like 2D memristors can act as artificial synapses [50]. This can help create neuromorphic circuits. They suggest that proteinoids might do the same by mimicking synaptic plasticity seen in their spontaneous oscillations. Moreover, Van De Burgt et al. noted that organic materials are flexible and cost-effective [51]. This means we could create proteinoid-based devices for wearable or implantable neuromorphic systems. These could be used in healthcare, like for real-time health monitoring or neural prosthetics.

### 4.4. Challenges in Integration with Conventional Electronics

Merging proteinoid assemblies with regular electronics has many tough challenges. We need to solve these problems for real-world use. Zikulnig et al. note that merging organic systems, like proteins, with silicon-based platforms presents challenges [52]. These include scalability, interface compatibility, and process integration. Proteinoid fabrication methods usually vary a lot from standard semiconductor techniques. Zhang et al. also note that van der Waals interfaces in 2D materials have poor quality [53]. They struggle with compatibility with silicon electronics. This issue is like proteinoid–device interfacing. Mismatched electrical properties can disrupt signal transduction. Müller et al. highlight challenges in integrating ferroelectric hafnium oxide memories with CMOS processes [54]. They mention material compatibility and process complexity. These issues may also occur with proteinoids, given their organic nature and sensitivity to harsh fabrication conditions. Li et al. highlight that memristor devices resemble protein-based spiking systems [55]. Yet, they face challenges integrating into standard electronic architectures. These issues arise from material inconsistencies and impedance mismatches. This suggests that proteinoids might encounter similar problems in achieving reliable connectivity. Thomas et al. discuss the precision needed to connect silicon quantum dot devices with custom electronics [56]. They suggest that protein–device coupling may need advanced methods. These could include custom intermediate layers or precise alignment techniques. These strategies are key for integrating functions in compact formats. These challenges highlight the need for new interfacing solutions. They help connect proteinoid assemblies with traditional electronics.

### 4.5. Implications for Prebiotic Evolution and Biomimetic Computing

Our findings greatly help us understand possible prebiotic pathways to early signal processing systems. Active structures can form from simple peptides. This suggests basic computing elements might have developed through self-assembly in early life settings. The observed differences in shapes support unique signal processing modes. This likely drove selection. This led to the evolution of more advanced information-handling capabilities.

GFD proteinoid systems are good options for biomimetic computing. The mix of simple structure, self-assembly properties, and complex electrical behaviour provides benefits over traditional semiconductors for some computing tasks. These systems are naturally variable and adaptable. This makes them great for tasks like pattern recognition and adaptive learning. They fit well in neuromorphic computing, where strict rules are less important than discovering new information.

The observed control of shape affects electrical properties. This suggests ways to engineer specific computing functions using directed self-assembly. One can create custom proteinoid networks by adjusting assembly conditions. This can help to favour certain shapes or arrangements at the interface. As a result, these networks may have unique signal processing traits. This method may create new biocompatible computing materials. These materials connect traditional electronics with biological systems.

### 4.6. Limitations and Future Directions

Our study shows clear links between shape and electrical behaviour in GFD proteinoid systems. However, we should note some limitations. First, the molecular mechanisms underlying the observed electrical phenomena remain incompletely characterized. Future studies using molecular dynamics simulations and spectroscopy could reveal how peptide arrangements and ion interactions create different electrical behaviours.

Next, we need to study how stable these structures are over time. We should also look into their electrical properties more closely. Studying how proteinoid assemblies change over time can help us understand their use in long-term computing.

The information processing abilities of these systems hint at their computational power, but we need better measurement. Setting standard measures for signal processing efficiency, information capacity, and computational complexity will help compare neuromorphic systems more effectively.

GFD proteinoid systems show promise, but challenges remain for their use in computing and biosensing. These include
Scaling issues from unpredictable assembly patterns at larger sizes;Stability problems in different environments;Difficulties in making reliable connections with standard electronic parts for real-world use.

These issues reflect challenges in cell-based biosensors. Environmental changes and problems with electronics make it hard to scale up and ensure reliability [57]. To address stability, insights from Shi et al. [58] suggest that PEGylated multifunctional peptides may boost GFD proteinoid strength. They achieve this by cutting down signal noise and improving performance in tricky biological settings. This could help to stabilize microspheres for biosensing uses. Mougkogiannis and Adamatzky et al. [59] point out that proteinoid microspheres create neural-like networks. However, these networks face problems with assembly and scaling. This makes it hard to achieve consistent performance when scaled up. Interface instability is a big problem in aptamer-modified quantum dot systems [60]. It makes it hard to transfer reliable signals from proteinoid systems to standard electronics. This issue needs better surface functionalization strategies. Graphene-based biosensors, as discussed by Menaa et al. [61], face similar scaling and environmental challenges. Their success in surface functionalization to stabilize signal transduction could be a helpful model for GFD systems. GFD proteinoid systems can solve today’s problems. They use strategies like peptide stabilization and custom surface changes. This approach allows for more reliable, scalable, and compatible platforms for future computing and biosensing.

In future studies, we plan to conduct systematic perturbation experiments across multiple dimensions. First, we will change specific amino acids. We will focus on glutamic and aspartic acid residues to adjust their charges. This will help us to control the electrostatic properties while keeping the overall shape intact. Next, we plan to create real-time imaging methods. We will use ion-selective fluorescent probes to see ion flux patterns. This will show how ions move during electrical activity in both microsphere and fibrillar assemblies. This approach will allow us to directly observe how specific morphological features facilitate or constrain ion movement pathways. Next, we will conduct studies with controlled electrical and mechanical changes. This will help us see how structural disruptions affect function. We aim to gather more proof of how structure impacts mechanisms. Environmental modulation experiments will help us find the key factors that affect both shape stability and electrical behavior. These factors include pH gradients, temperature shifts, and ionic strength variations.

Future research should explore more peptide compositions. This will help expand the range of morphologies and electrical properties available. Studying how temperature, pH, and ionic composition affect structure and electrical behaviour will help us understand how these systems adapt. Also, creating links between proteinoid assemblies and regular electronic parts could lead to hybrid computers. These would use the special features of both systems.

## 5. Conclusions

We found that Glu-Phe-Asp proteinoid assemblies show a strong link between their shape and electrical behaviour. Microspheres create quick, neuron-like spikes. Fibers show slow, steady changes. Heterogeneous mixtures have unique traits that hint at early information processing capabilities. These findings show new ways that prebiotic pathways can influence signal-processing systems. They also suggest innovative methods for biomimetic computing. Simple self-assembling proteinoids can create different electrical behaviours. This shows how nature builds complexity using basic physicochemical principles. This idea connects from tiny molecules to larger biological systems.

## Figures and Tables

**Figure 1 biomimetics-10-00360-f001:**
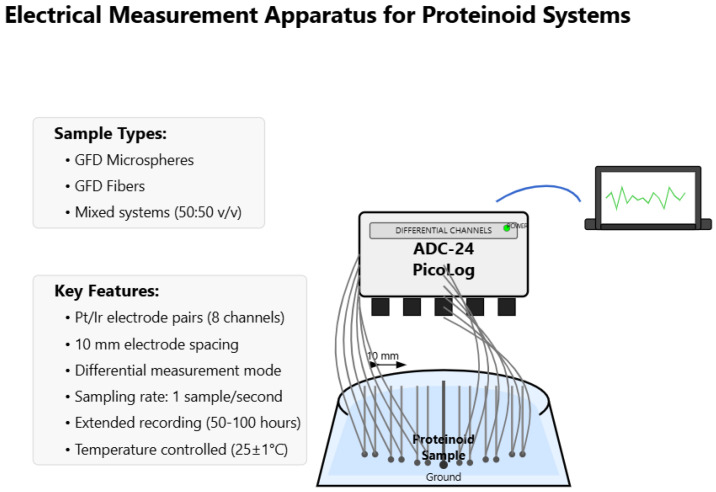
A diagram of the electrical measurement tool for recording extracellular potentials from proteinoid assemblies. The setup has a glass vial with rehydrated proteinoid samples. These samples can be microspheres, fibers, or a 50:50 *v*/*v* mixture. There are eight pairs of Pt/Ir electrodes placed 10 mm apart. Electrodes are connected to an ADC-24 PicoLog data logger operating in differential mode with a sampling rate of 1 sample per second. The central ground electrode provides a common reference point for all measurements. This setup allows for 50 to 100 h of recording at a stable temperature of 25 ± 1 °C. It captures quick electrical events and shows long-term patterns.

**Figure 2 biomimetics-10-00360-f002:**
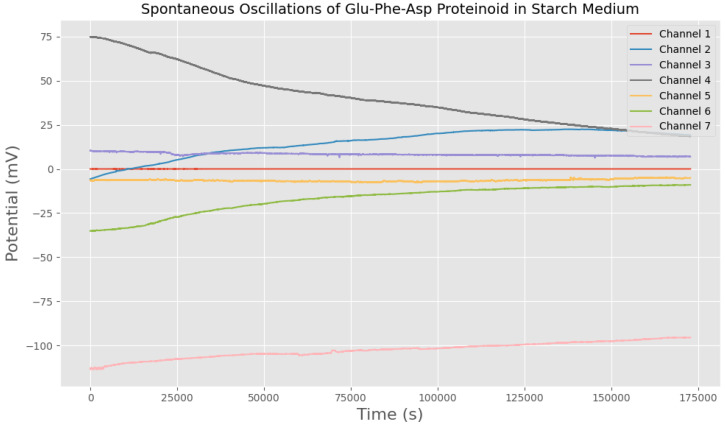
Changes in electrical potentials of Glu-Phe-Asp (GFD) proteinoid microspheres in starch medium observed over about 48 h (175,000 s). The starch medium changes the electrical response seen in pure GFD microsphere preparations. Instead of sharp spikes, it causes gradual drift patterns across several recording channels. Channel 4 (gray) shows a pronounced decline from +75 mV to +20 mV, while Channel 7 (pink) maintains a strong negative potential around −100 mV throughout the recording. The lack of action potential spikes and burst patterns in this control experiment shows that the electrical behaviors in pure GFD systems are true traits of the proteinoid structures. They are not just measurement errors. This control shows how the surrounding medium can change the electrochemical gradients created by GFD assemblies.

**Figure 3 biomimetics-10-00360-f003:**
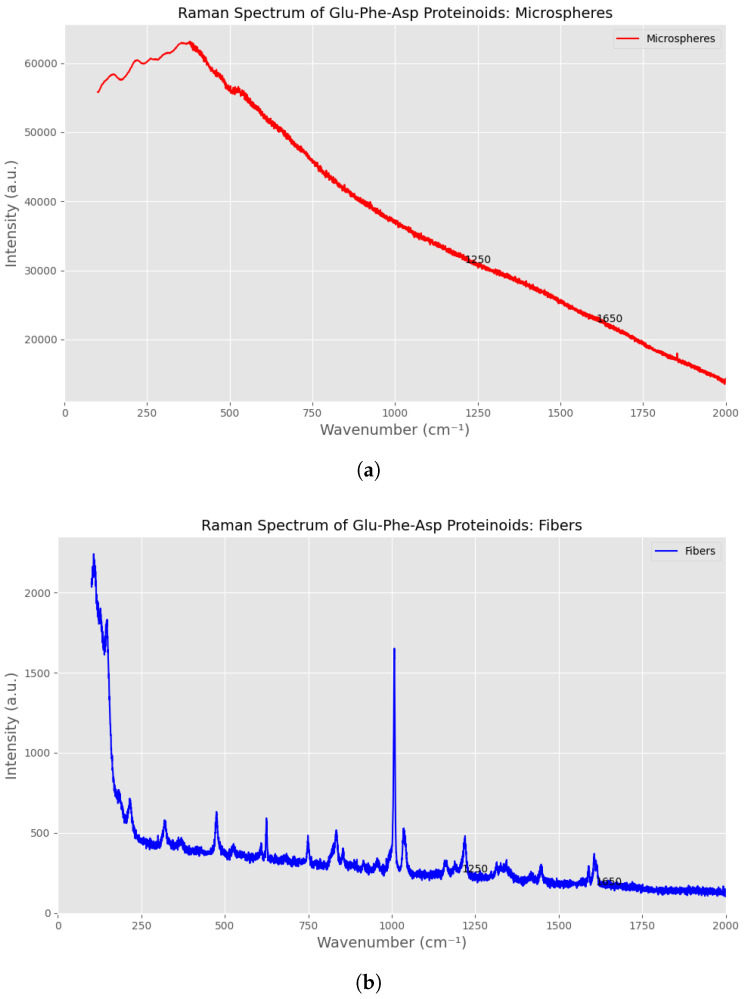
Raman spectra display intensity as a function of wavenumber (cm^−1^) for the (**a**) microspheres, (**b**) fibers, and (**c**) the fibers and microspheres composite. The spectral range spans from 100 to 1999 cm^−1^. Key vibrational modes are observed at 1250 cm^−1^ (amide III) and 1650 cm^−1^ (amide I), corresponding to characteristic features of the proteinoid backbone structure. Microspheres exhibit broader and more intense peaks, indicating a higher degree of molecular disorder. In contrast, fibers show sharper peaks, reflecting a more ordered molecular arrangement. The composite spectrum presents a combination of both morphologies, suggesting that this structural blend may enhance the material’s applicability in biomaterials where a balance of order and flexibility is beneficial.

**Figure 4 biomimetics-10-00360-f004:**
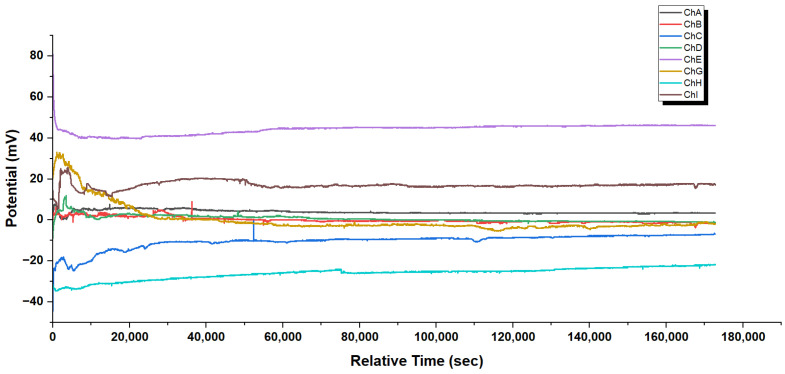
Temporal changes in extracellular potentials recorded from a mixture of Glu-Phe-Asp (GFD) microspheres and fibers in a 50:50 *v*/*v* ratio. Multiple recording channels (ChA–ChI) reveal distinct electrophysiological profiles across the sample. The system shows initial changes from 0 to 10,000 s. It stabilizes over time. The steady-state potentials range from −40 mV to +45 mV. Channel E (purple) has a steady positive potential of about +45 mV. In contrast, Channel H (light blue) stabilizes at around −25 mV. Intermediate channels show different equilibration patterns. Some channels, like ChI (brown), have metastable plateaus around +15 mV. This varied electrical behaviour hints at different functional areas in the microsphere–fiber network. These areas may link to the shapes of the fibrillar and spherical structures. The long-term stability observed (over 60,000 s) shows that strong electrochemical gradients form in the mixed GFD assembly. This is typical of the extracellular field potentials created by self-assembling peptide structures.

**Figure 5 biomimetics-10-00360-f005:**
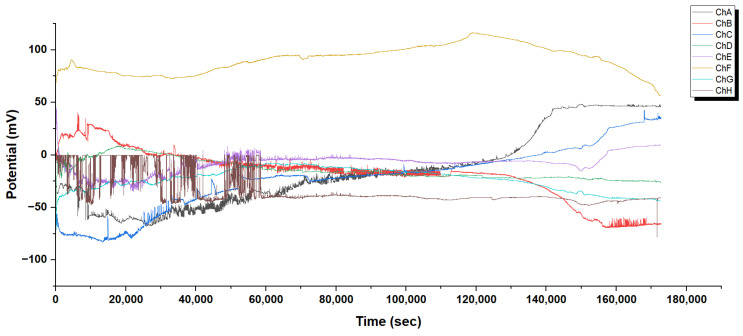
Long-term recordings show electrical activity from pure Glu-Phe-Asp (GFD) microspheres. They display complex spiking dynamics over about 50 h, or 180,000 s. The recording shows three clear phases of electrophysiological behaviour. In the initial phase (0–40,000 s), high-frequency spiking occurs, especially in channels B, E, and G. Spike amplitudes range from 5–15 mV, and bursts have distinct patterns. In the transitional phase, (40,000–120,000 s), there is a gradual change in potential trajectories. Spiking frequency decreases during this time. In the late phase (120,000–180,000 s), channels spread across a wide potential range of −70 to +110 mV. Channel-specific spiking patterns, especially in channel B at −70 mV, become noticeable. Channel F (yellow) stays at high potentials during the recording, peaking at +110 mV. Other channels, however, show shifts between positive and negative values. This spontaneous spiking behaviour shows a key difference from GFD fiber structures. GFD fibers have smooth, steady potential changes and do not show action potential-like events. Spikes appear only in microsphere preparations. Their round shape helps create the electrochemical compartments. This shape allows for quick ionic flows across their surfaces. In contrast, fibers have a long, non-compartmentalized structure. This shape prevents the fast potential changes that spikes need to form.

**Figure 6 biomimetics-10-00360-f006:**
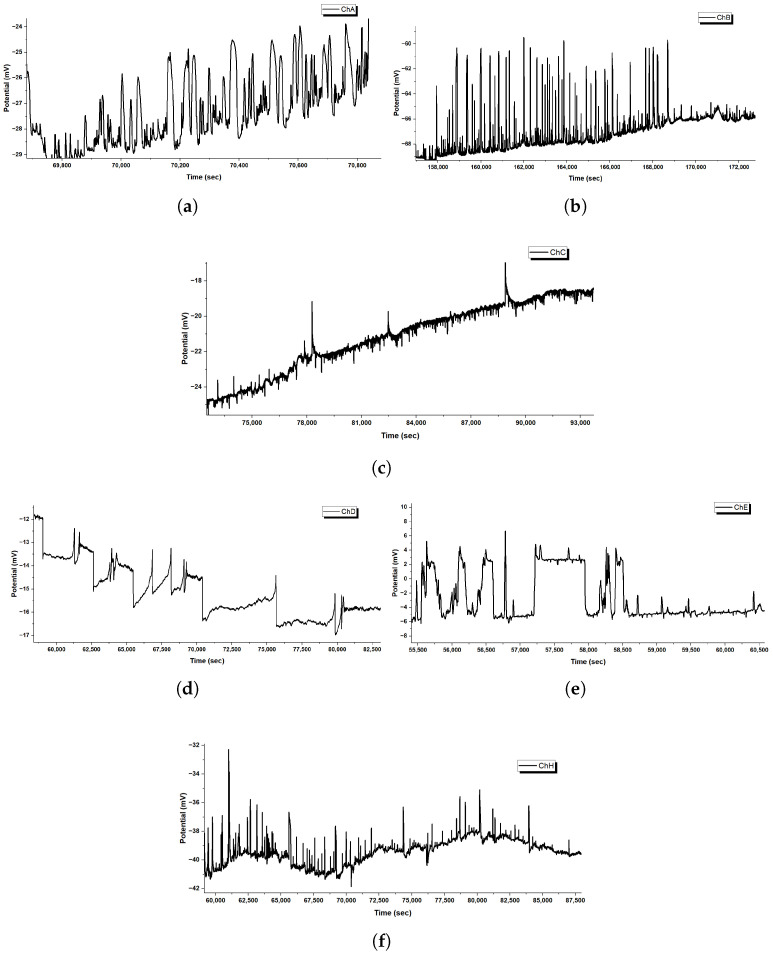
We conduct a high-resolution analysis of spike waveforms in Glu-Phe-Asp microsphere networks. This analysis spans multiple recording channels. (**a**) Channel A shows quick upward spikes of 2–4 mV on a slowly rising baseline of −29 to −24 mV. The spikes become more common and larger as the recording continues. (**b**) Channel B shows clear, downward spikes with an amplitude of 2–3 mV. After 170,000 s, it shifts to a steadier baseline with fewer spikes. (**c**) Channel C displays clear, large spikes (3–5 mV) that stand out from a slowly rising baseline (−25 to −19 mV). The waveforms have a quick rise and a gradual decay. (**d**) Channel D shows step-like changes between stable potential states (−13 to −17 mV). There are sudden shifts, followed by flat periods, which indicate clear state transitions. (**e**) Channel E shows complex two-way spiking. It has large swings, reaching 8–10 mV. It also features long plateaus at various potential levels. This behaviour indicates bistable switching. (**f**) Channel H displays high-frequency spikes with small amplitudes (1–2 mV). These spikes sit on a fluctuating baseline that ranges from −42 to −36 mV. Additionally, there are periodic bursts. These different spiking patterns are very unlike the steady potential changes in fibrillar structures. This shows how the shape of microspheres creates separate areas that can quickly move ions across boundaries. This movement leads to action potential-like events, featuring distinct depolarization and repolarization phases. The differences in spike shape across channels show that there are local variations in electrochemical properties in the microsphere network. This might allow for complex information processing through the integration of signals over space and time.

**Figure 7 biomimetics-10-00360-f007:**
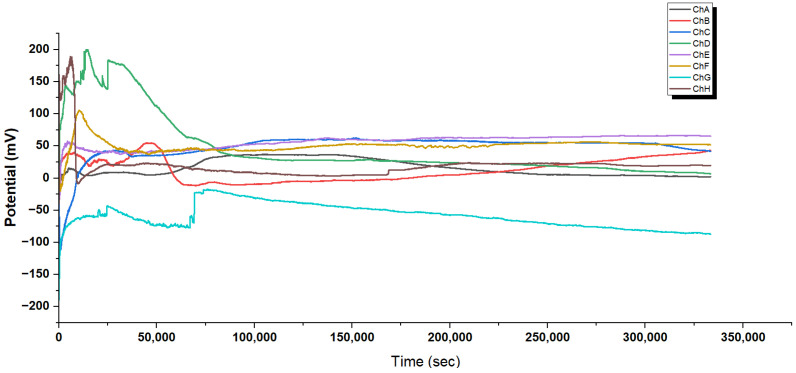
We recorded extracellular potentials from pure Glu-Phe-Asp (GFD) fiber networks for about 100 h, or 360,000 s. These long-term recordings showed unique electrophysiological dynamics without classical spiking activity. The temporal evolution shows three main phases. In the initial high-amplitude transient phase (0–50,000 s), there are large potential changes. Channels D and F reach peaks of +200 mV and +100 mV. In the transition phase (50,000–150,000 s), channels start to align their potential positions. In the extended equilibrium phase (150,000–360,000 s), stability is key, with very few fluctuations. GFD fibers do not show rapid spike events like microsphere preparations do. Instead, they have smooth, gradual potential changes. This matches the distributed ionic gradients found in their long fiber structures. The recordings show clear polarization bands. Channels split into three groups, as follows. Positive: Channels C, E, F (+40 to +65 mV). Neutral: Channels A, B, H (−10 to +30 mV). Negative: Channel G (−60 to −90 mV) This separation of potentials shows that the fiber network has different functions, even though it looks the same. The stability of these distributions lasts over 200,000 s. This shows that strong electrochemical gradients form and persist. This is due to the fibrillar structure. These gradients can store information long-term. This is different from the dynamic signals in microsphere systems.

**Figure 8 biomimetics-10-00360-f008:**
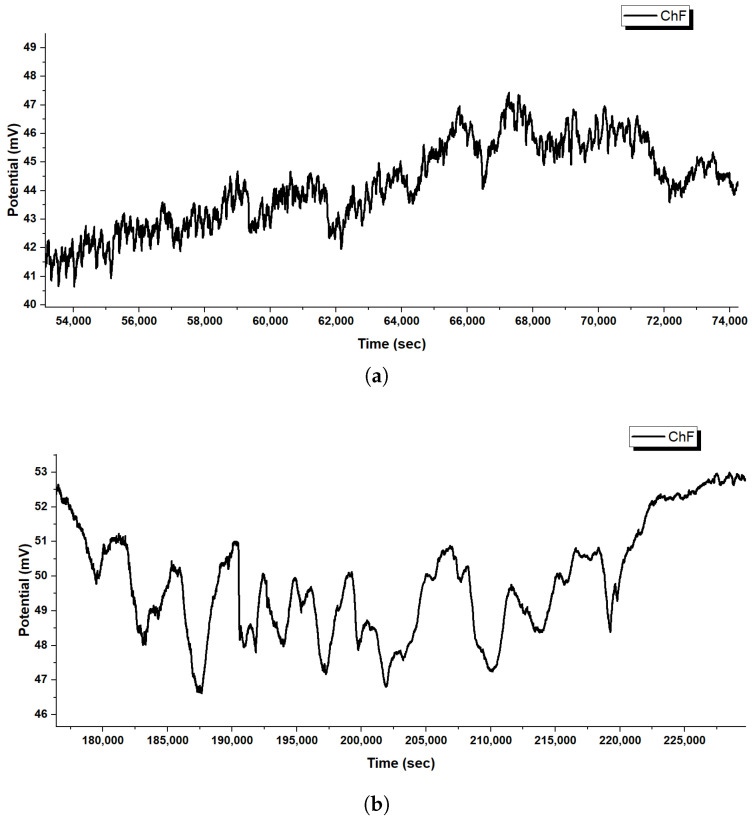
High-resolution temporal analysis of Glu-Phe-Asp fiber network activity from Channel F recordings at different time points. (**a**) Early recording (54,000–74,000 s) shows a gradual rise in potential. It rises from about 41 mV to 47 mV. You can see micro-oscillations with amplitudes of 0.5–1.0 mV. There are no clear spiking events. The steady rise (ΔV≈+6 mV over 20,000 s) shows a gradual buildup of charge in the fiber structure. The small-amplitude fluctuations reflect ionic movements spread out across the area, not isolated discharge events. (**b**) Middle-phase recording (180,000–225,000 s) shows new quasi-periodic oscillations. These oscillations have peak-to-peak amplitudes of 3–5 mV. The periodicity varies, with major troughs occurring about every 5000–10,000 s. Slow-wave oscillations keep baseline potentials between 46 and 53 mV. They have uneven waveforms. The phases descend at a gradual pace and then rise rapidly. These fiber-specific oscillations occur over longer times than the quick spikes seen in microsphere preparations. This points to different mechanisms that control changes in fibrillar and spherical shapes. The long-lasting features of these oscillations suggest that ions move in large groups. This happens across the fiber network, not in small, quick bursts like in compartmentalized structures.

**Figure 10 biomimetics-10-00360-f010:**
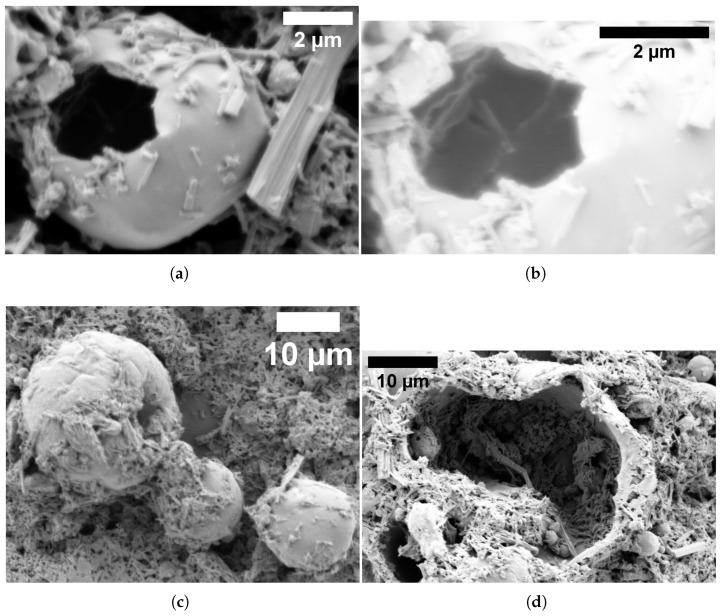
SEM characterization of heterogeneous Glu-Phe-Asp (GFD) assemblies revealing microsphere–fiber interactions and morphological diversity. (**a**) Perforated microsphere (d=6.576μm) with central aperture ($d_{hole}=3.047\;\mu$m) in direct contact with elongated fibrillar structure (length = 5.930μm). Structural integration at the microsphere–fiber interface suggests possible electrochemical coupling between various domains. This was observed using an ETD detector with HV set to 2.00 kV, magnification at 35,125×, HFW of 11.8μm, and pressure at $1.40\times 10^{-6}$ Torr. (**b**) High-resolution images show a microsphere with an uneven edge near fibrillar elements. This setup highlights the localized surface interactions at the shape’s boundary. The clear differences in electron density show varied compositional organization. This was observed using an ETD detector at HV = 2.00 kV, with a magnification of 63,039×, an HFW of 6.57μm, and a pressure of $1.40\times 10^{-6}$ Torr. (**c**) The multi-generational microsphere assembly shows a size range with different sizes: d=25.373μm, 14.156μm, 14.869μm, and 2.329μm. It also shows signs of budding-like growth. The spatial arrangement shows how the microsphere population develops. This was observed using an ETD detector at HV = 2.00 kV, with 4297× magnification and an HFW of 96.4μm. (**d**) The advanced-stage hollow microsphere structure (33.152 μm) has smaller microspheres inside (1.5 μm, 4.5 μm). This shows a rare encapsulation phenomenon in synthetic self-assembling systems. The hierarchical setup has nested substructures. This design allows for compartmentalization of electrochemical processes. For example, consider the ETD detector. It operates at HV = 2.00 kV with a magnification of 8052×. The HFW is 51.5μm, and the pressure is $1.22\times 10^{-6}$ Torr.

**Figure 11 biomimetics-10-00360-f011:**
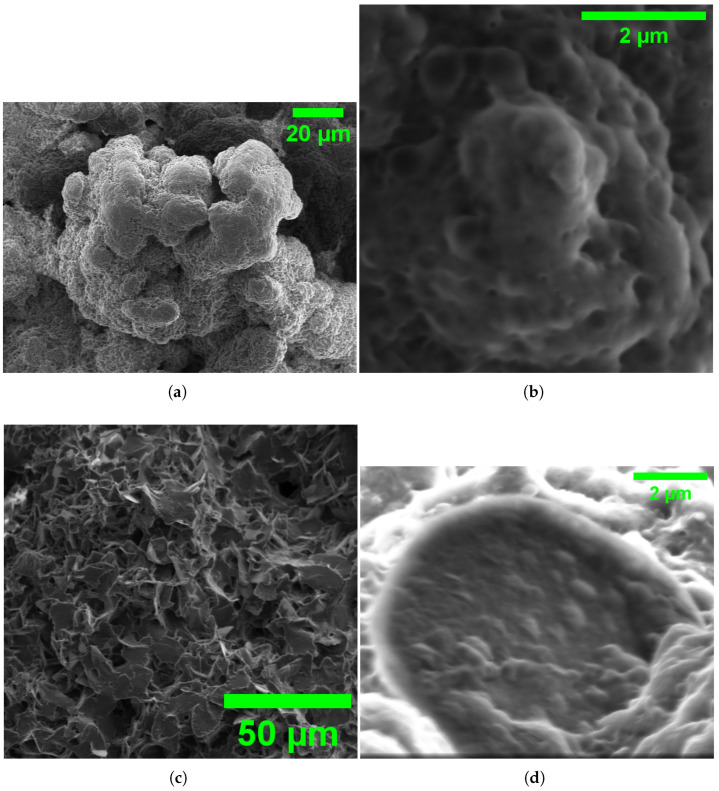
Scanning electron microscopy (SEM) images of L-ASP morphology in water: (**a**) overview of aggregated L-ASP structures at 20 μm scale, displaying a rough, clustered surface texture; (**b**) close-up view at 20 μm scale, revealing complex surface details and porous features; (**c**) larger-scale view at 50 μm, showing the distribution and uniformity of L-ASP particles; and (**d**) high-magnification image at 2 μm, highlighting the fine nanoscale surface morphology and structural details of individual L-ASP particles.

**Figure 12 biomimetics-10-00360-f012:**
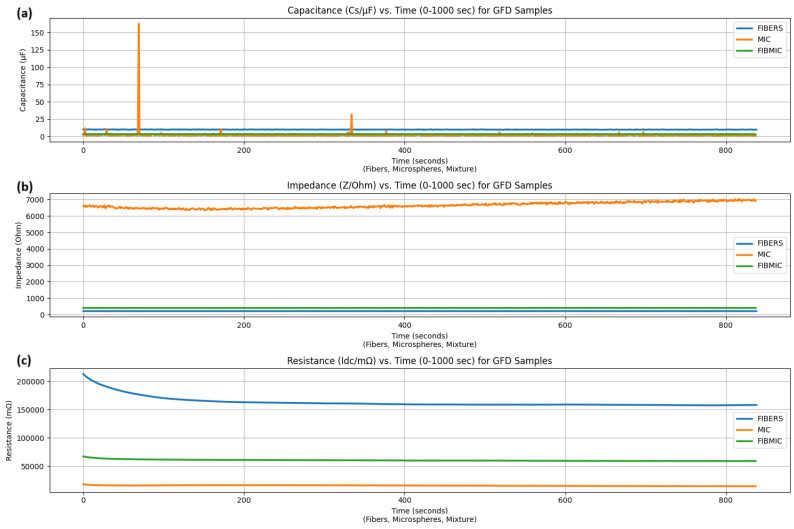
Changes in electrical properties of different GFD proteinoid structures over 1000 s. Capacitance (Cs/μF) measurements show that fibrillar structures have much higher baseline values. The values are 9.912±0.171
μF for fibrillar (blue), 1.926±5.735
μF for microspheres (orange), and 3.328±0.076
μF for mixed morphologies (green). Microspheres show distinct transient capacitance spikes, reaching up to 162.514 μF. This indicates quick charge buildup events. (**b**) Impedance (Z/Ohm) profiles show clear differences between morphologies. Microspheres have much higher impedance at 6646.282±178.664 Ohm. In contrast, fibers measure 209.400±0.286 Ohm, while mixed systems are at 404.235±1.091 Ohm. (**c**) Resistance (Idc/mΩ) measurements show an inverse relationship with capacitance. Fibers have the highest resistance at 163,067.613 ± 9253.064 mΩ. Next are mixed structures at 60,424.487 ± 1293.986 mΩ. Finally, microspheres measure 15,830.739 ± 652.514 mΩ. These unique electrical signatures relate to structure. Fibrillar shapes help spread charge but limit current flow. Microspheres allow ionic transport and keep high impedance. Mixed systems show traits in between, hinting at functional synergy among different structures.

**Figure 13 biomimetics-10-00360-f013:**
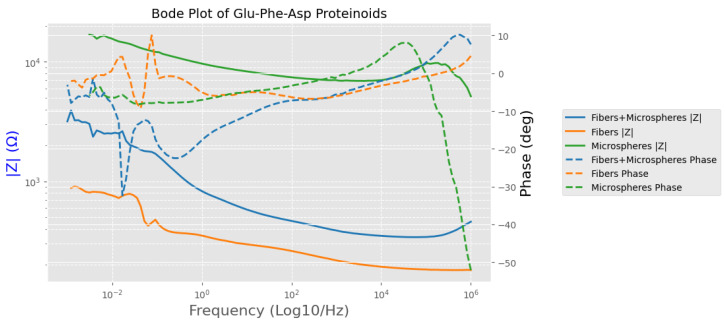
Bode plot of Glu-Phe-Asp proteinoids showing impedance magnitude |Z| (left axis, log scale) and phase (right axis) against frequency (log scale). This data is for fibers, microspheres, and the fibers and microspheres composite. Fibers show the lowest impedance, reaching |Z| as low as $10^{2}$
Ω at high frequencies. This suggests they mainly behave resistively. This effect likely comes from better charge transport along their fibrous structures. Microspheres have the highest impedance, reaching |Z| values up to $10^{4}$
Ω. They also show a phase shift near $-50^{\circ }$ at $10^{6}$ Hz. This indicates a strong capacitive effect, likely from ionic interactions on their surface. The fibers and microspheres composite displays intermediate |Z| values, reflecting a balanced resistive–capacitive response. These trends show that fibers might work like a simple resistor. In contrast, microspheres probably fit a parallel RC circuit. This highlights their potential for custom uses in bioelectronics.

**Figure 14 biomimetics-10-00360-f014:**
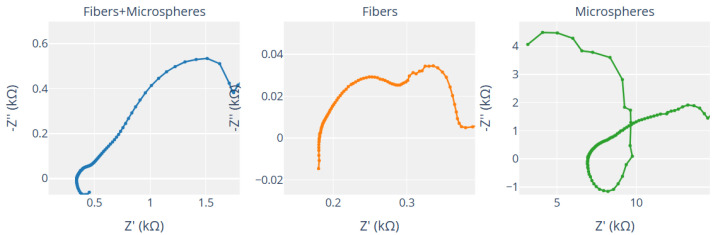
Nyquist plots showing Glu-Phe-Asp proteinoids. They compare real impedance $Z^{\prime }$ with negative imaginary impedance $-Z^{\prime \prime }$. The plots include (**a**) fibers and microspheres, (**b**) fibers, and (**c**) microspheres. Fibers show a compressed semicircle. The real part, $Z^{\prime }$, ranges from 0.180 to 0.181 kΩ. The imaginary part, $-Z^{\prime \prime }$, goes from −0.015 to −0.008 kΩ. This suggests a mainly resistive response. The charge transfer resistance is low, likely due to good ionic conduction along the fibers. Microspheres show a larger semicircle with $Z^{\prime }$ ranging from 3.162 to 5.024 kΩ. The $-Z^{\prime \prime }$ values go from −4.492 to −4.064 kΩ. This indicates strong capacitive impedance, like a parallel RC circuit. It could be caused by ionic interactions on the surface. The fibers + microspheres composite has a mid-range response. Its $Z^{\prime }$ varies from 0.418 to 0.455 kΩ, while $-Z^{\prime \prime }$ ranges from −0.071 to −0.061 kΩ. This indicates balanced resistive–capacitive behavior. One can zoom into specific regions of interest, such as fibers and microspheres ($Z^{\prime }$: 0.400 to 0.450 kΩ, $-Z^{\prime \prime }$: −0.080 to −0.040 kΩ), fibers ($Z^{\prime }$: 0.180 to 0.182 kΩ, $-Z^{\prime \prime }$: −0.015 to −0.005 kΩ), and microspheres ($Z^{\prime }$: 3.0 to 3.5 kΩ, $-Z^{\prime \prime }$: −4.5 to −3.5 kΩ), to explore detailed electrochemical dynamics for potential bioelectronic applications.

**Figure 15 biomimetics-10-00360-f015:**
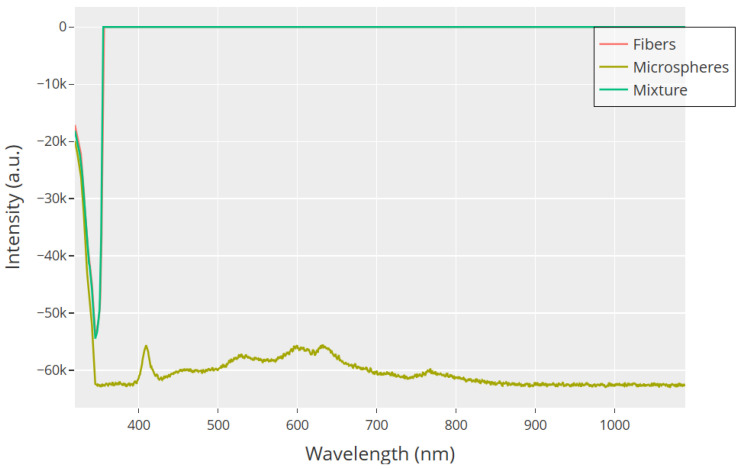
UV–Visible absorption spectra from 350 to 1050 nm show unique optical behaviors in GFD proteinoid morphologies. Microspheres (olive) keep their absorption profile, showing peaks at 410 nm and broad features from 500 to 650 nm. In contrast, fibers (red) and their mix with microspheres (teal) show surprising optical changes. The initial steep absorption edge at 350–375 nm represents π–$\pi^{*}$ transitions in aromatic phenylalanine residues. The mixture makes a big change around 375 nm. It shifts from high absorption to complete transparency in the visible and near-infrared range. This strange optical behavior hints at complex interactions between light and matter in the mixed assembly. It may be linked to constructive and destructive interference at the interfaces of the microspheres and fibers. The sharp change also backs our idea. It shows that mixed shapes lead to unique electrical behaviors. This happens through cooperation among structural elements.

**Figure 16 biomimetics-10-00360-f016:**
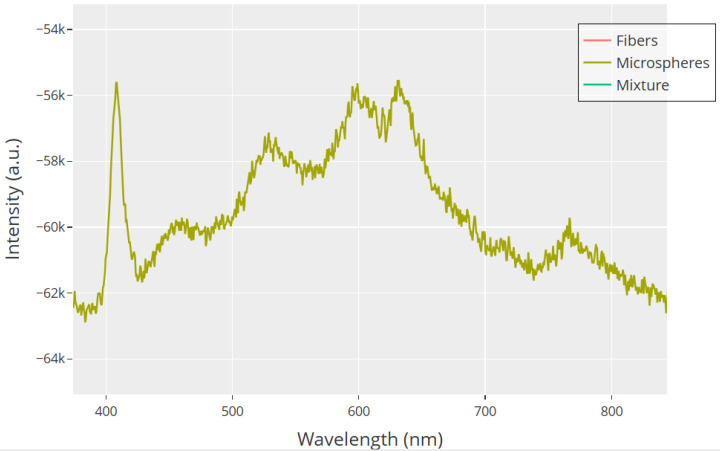
UV–Visible absorption spectra of Glu-Phe-Asp (GFD) proteinoid assemblies show clear spectroscopic signatures. These signatures vary across different morphologies in the 380–850 nm range. Microspheres (olive) show a complex spectrum with several absorption bands. There is a sharp peak at 410 nm. This peak likely comes from n–$\pi^{*}$ transitions of carbonyl groups. Broad absorption occurs between 570–630 nm, linked to electronic transitions in the peptide backbone. Additionally, a smaller feature at 750 nm may relate to extended conjugation networks. Fibers (red) and the 50:50 mixture (teal) have very different profiles. They overlap at lower wavelengths but diverge a lot past 400 nm. The mixture shows a flat response, which suggests aggregation-induced spectral damping. These unique spectral signatures match the various molecular patterns seen in SEM analysis. They also back up the structure-dependent electrical behaviors noted in our recordings.

**Figure 17 biomimetics-10-00360-f017:**
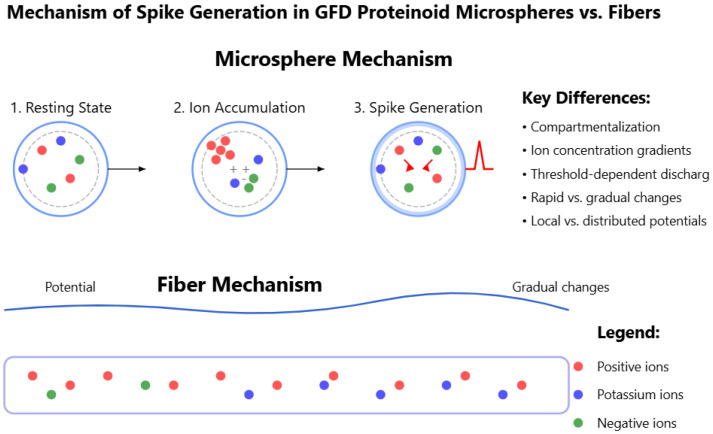
Diagram showing how spike generation works in Glu–Phe–Asp (GFD) proteinoid microspheres and fibers, highlighting how their shapes affect their electrical behavior. The microsphere mechanism has three stages: the resting state, where ions are balanced; ion accumulation, which creates concentration gradients; and spike generation, where a threshold triggers discharge. The fiber mechanism shows gradual potential changes due to distributed ion movement. The key differences are compartmentalization, ion gradients, rapid versus gradual changes, and local versus distributed potentials. Positive ions are shown in red, potassium ions in blue, and negative ions in green. This matches the quick spikes seen in Figure 6 and the slow waves in Figure 8, demonstrating how structure affects GFD network dynamics.

**Table 1 biomimetics-10-00360-t001:** Representative examples of protein polymorphism across structural and functional categories.

Protein Type	Polymorphic Structures	Functional Implications
Amyloidogenic Proteins [19]	Soluble monomers, oligomers, protofibrils, mature fibrils	Transition from physiological function to pathological aggregates in neurodegenerative disorders (Alzheimer’s, Parkinson’s)
Prion Proteins [20]	${\mathrm{PrP}}^{C}$ (cellular form), ${\mathrm{PrP}}^{\mathrm{Sc}}$ (scrapie form)	Conformational change from predominantly α-helical to β-sheet rich structure leads to infectious propagation and neurodegeneration
Hemoglobin [21]	Tense (T) state, relaxed (R) state	Allosteric regulation of oxygen binding affinity through quaternary structural transitions
Heat Shock Proteins [22]	Monomeric, oligomeric, and substrate-bound forms	Chaperone activity modulated by dynamic assembly/disassembly in response to cellular stress conditions
Crystallins [23]	Soluble oligomers, insoluble aggregates, amorphous deposits	Age-related transition from transparent lens proteins to cataract-forming structures
Viral Capsid Proteins [24]	Pentameric assemblies, hexameric assemblies, mature capsid	Structural transitions essential for viral assembly, maturation, and host cell interaction
Proteinoids [this work]	Microspheres, fibers, mixed morphologies	Self-assembly into distinct structures with differential electrical properties and information processing capabilities
Intrinsically Disordered Proteins [25]	Extended random coils, partially structured intermediates, folded states	Functional plasticity enabling interactions with multiple binding partners in signaling pathways
G-Protein Coupled Receptors [26]	Inactive conformation, active conformation, various intermediate states	Conformational selection determines signaling outcomes in response to different ligands
Tubulin [27]	α/β heterodimers, protofilaments, microtubules, sheets	Dynamic instability and structural transitions essential for cellular division and intracellular transport

**Table 2 biomimetics-10-00360-t002:** Key Raman spectral features for Glu-Phe-Asp proteinoid morphologies. Fibers exhibit numerous sharp, well-defined peaks throughout the spectrum, particularly at 1000 cm^−1^ (strong, narrow peak) and 1250 cm^−1^ (amide III), indicating high molecular order. In contrast, microspheres show a smooth profile with minimal distinct peaks and a broad, continuous decline from 500–2000 cm^−1^, suggesting molecular disorder. The fiber+microsphere mixture displays intermediate characteristics with some defined features at 750 cm^−1^ and 1250 cm^−1^ superimposed on a rising baseline.

Morphology	Key Spectral Features	Structural Implications
Fibers	Multiple sharp peaks (250–1750 cm^−1^) Prominent peaks at ∼500, 650, 1000, 1250, 1650 cm^−1^ Sharp, high-intensity peak at 1000 cm^−1^	High degree of molecular order Consistent inter-chain interactions Aligned aromatic (Phe) residues Directional hydrogen bonding
Microspheres	Smooth spectral profile Broad peak at ∼350 cm^−1^ Continuous intensity decline from 500–2000 cm^−1^ Minimal distinct features at 1250, 1650 cm^−1^	Significant molecular disorder Heterogeneous molecular arrangement Isotropic self-assembly Random orientation of side chains
Fibers + Microspheres	Intermediate spectral profile Step-like increase at ∼750 cm^−1^ Subtle features at 1250, 1650 cm^−1^ Rising baseline beyond 1700 cm^−1^	Combined structural elements Dominated by microsphere contributions Mixed ordered/disordered regions Complex interfacial interactions

**Table 3 biomimetics-10-00360-t003:** Detailed statistics of impedance (Z/Ohm), capacitance (Cs/μF), and resistance (Idc/mΩ) for GFD proteinoid samples over 1000 s, derived from time-series data.

	Impedance (Z/Ohm)					Capacitance (Cs/μF)					Resistance (Idc/mΩ)					
Sample	Mean	Std	Min	Max	Median	Mean	Std	Min	Max	Median	Mean	Std	Min	Max	Median	Slope
Fibers	209.400	0.286	208.349	210.225	209.401	9.912	0.171	9.436	10.434	9.898	163,067.613	9253.064	157,464.172	212,725.067	159,023.422	−26.613
Mixture	404.235	1.091	400.544	408.045	404.226	3.328	0.076	3.062	3.601	3.326	60,424.487	1293.986	59,054.138	67,298.653	59,998.276	−4.707
Microspheres	6646.282	178.664	6346.610	7038.241	6620.197	1.926	5.735	0.633	162.514	1.452	15,830.739	652.514	14,643.011	18,024.274	16,000.094	−2.483

## Data Availability

The datasets generated and analysed during the current study are available in the zenodo repository, which can be accessed through the following web address: [https://zenodo.org/records/15044285 (accessed on 15 May 2025)].

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
