# Peer review of "Polymorphism in Glu-Phe-Asp Proteinoids"

_biomimetics, 2025, doi:10.3390/biomimetics10060360_

Round 1

Reviewer 1 Report

Comments and Suggestions for Authors

Polymorphism in Glu-Phe-Asp Proteinoids

  1. Lacks concise statements of methods and specific findings or contributions. Could be strengthened with clearer aims and summarization of the paper’s key arguments.
  2. The objectives of the article are implied but not clearly defined.

  3. Introduction Needs a more structured breakdown of how the article will proceed.

  4. Some concepts (like neuromuscular control) are introduced briefly and could benefit from deeper elaboration.

  5. Limited discussion on limitations or challenges (e.g., scalability, stability, real-world applications).

Reviewer 2 Report

Comments and Suggestions for Authors

This manuscript presents an interesting study exploring the relationship between the structural and electrophysiological polymorphism of Glu-Phe-Asp (GFD) proteinoids. The authors show that GFD assemblies exhibit structure-dependent electrical behavior. The study is generally well-organized, combining morphological characterization via SEM with electrophysiological recordings. The authors make insightful connections between structure and function, suggesting that morphological polymorphism may underlie complex dynamic behaviors in simple peptide systems. However, the issues listed below should be addressed before publication to strengthen the study.

Major Comments

1. The manuscript lacks control experiments to exclude the possibility that the observed electrical behaviors arise from the background solution environment or measurement setup. I recommend including control experiments using, such as pure water/buffer, non-polarized GFD mixture, or mechanically disrupted GFD assemblies.

2. The authors propose mechanisms relating morphology to ion movement and electrical behavior, but there is no direct chemical analysis confirming these mechanisms. Including spectroscopic characterization of the proteinoids to assess crosslinking, charge states, or structural features, using FTIR, Raman, or NMR, would strengthen the study.

3. The long-term stability of the proteinoid structures (or structural changes) is not assessed.

4. It would be valuable to conduct accelerated aging or prolonged incubation studies to test whether the observed electrophysiological behavior persists, deteriorates, or evolves over time.

5. The SEM imaging reveals considerable variability in microsphere size and fiber organization. However, no statistical analysis, such as particle size or fiber length distribution, is provided. Including quantitative analysis of morphologies would offer a more rigorous quantitative assessment of the assembly process and its reproducibility.

Minor Comments

1. No figure caption for Figure 7 (d)

2. The authors briefly mention neuromorphic computing but could further expand on potential applications and challenges, especially regarding integrating proteinoid assemblies with conventional electronics.

3. Increasing the font size in some Figures 4 and 9 would improve readability.

4. Adjust Figure 10 to avoid the overlap between the gradual line and the text.

Reviewer 3 Report

Comments and Suggestions for Authors

This manuscript entitled " Polymorphism in Glu-Phe-Asp Proteinoids" presents an intriguing study on polymorphic assemblies derived from Glu-Phe-Asp (GFD) proteinoids, exploring their structural and electrochemical properties. The investigation into how morphology influences electrical behavior in synthetic proteinoid systems is timely and potentially impactful, particularly for biomimetic materials applications. However, the manuscript has several critical deficiencies and must be majorly revised prior publication.

  1. This manuscript investigates how structural polymorphism influences electrical properties in proteinoid systems, a promising and innovative area. However, the authors do not sufficiently clarify how their work significantly advances beyond existing literature. The novelty and originality of the presented findings must be explicitly articulated in the introduction and discussion sections. The authors should clearly position this study within the broader context of biomimetic materials research, highlighting precisely what new insights or capabilities their work provides compared to published
  2. This manuscript lacks sufficient detailsregarding the molecular-level design principles that govern the formation and stabilization of microspheres, fibers, and mixed morphologies. Although SEM images effectively demonstrate morphological diversity, there is minimal discussion on the molecular interactions (e.g., hydrogen bonding, π-π stacking, hydrophobic interactions) responsible for structural polymorphism. The authors must include comprehensive spectroscopic or diffraction-based characterization (e.g., FT-IR, Circular Dichroism, XRD) to elucidate the molecular interactions and secondary structural motifs underpinning the observed morphologies. Without this information, the mechanistic insights remain speculative and incomplete.
  3. The criteria for distinguishing between “microsphere,” “fiber,” and “mixed” morphologies should be quantitatively defined, possibly supported by image analysis or statistical metrics. The authors must include comprehensive characterization, such as DLS (Dynamic Light Scattering), SAXS (Small Angle X-ray Scattering), and Raman spectroscopic.
  4. The authors describe distinct preparation methods for microspheres and fibers but do not adequately explain how these methods influence the resulting morphologies at a molecular level. For example, this manuscript should clearly articulate how the thermal condensation conditions (temperature, reaction time, cooling rates, solvent effects) specifically drive the formation of either spherical or fibrillar assemblies. Detailed kinetic studies or controlled variation experiments would significantly strengthen the manuscript by clarifying the assembly pathways and providing mechanistic insights.
  5. The connection between morphology and functionshould be deeply investigated. Additional experiments (e.g., perturbation of assembly, real-time imaging of ion flux, or chemical modification) would help establish causality.
  6. The electrochemical measurements presented (impedance, capacitance, resistance) are intriguing but currently lack sufficient depth and clarity. The authors must provide clearer justification for the choice of measurement conditions (frequency range, electrolyte conditions, electrode setup). Additionally, the manuscript would greatly benefit from systematic electrochemical impedance spectroscopy (EIS) analyses, including Nyquist and Bode plots, to better interpret charge storage mechanisms, ion transport kinetics, and interface properties. Furthermore, the authors should discuss in detail how the molecular-scale features of each morphology directly correlate with the observed electrochemical properties.
  7. The authors' interpretation of microsphere electrical spikes as "action-potential-like" events is overly speculative and requires careful qualification.Biological action potentials are driven by highly specific ion channels and membrane structures. The authors must clarify how the synthetic proteinoid structures achieve similar transient voltage behaviors without such specialized biological machinery.
  8. The mechanistic models proposed (e.g., Hodgkin-Huxley-like frameworks) are only qualitatively described. Quantitative fitting or simulation results would greatly strengthen the claims.Besides, a more rigorous physicochemical model or computational simulation (e.g., molecular dynamics or continuum electrostatic modeling) is required to substantiate these claims and provide a scientifically sound explanation of the observed electrical phenomena.
  9. The manuscript does not adequately address the long-term stability and reproducibility of the proteinoid assemblies and their electrical behaviors. The authors should provide data on the stability of these structures under physiological conditions (e.g., pH, ionic strength, temperature variations) and discuss batch-to-batch reproducibility. This information is crucial for evaluating the practical applicability and reliability of these systems as biomimetic materials or computing elements.
  10. Adequate control experiments are missing. For instance, comparisons with simpler peptide assemblies or known reference materials would help validate the claimed unique electrical behaviors.

Other concerns following:

Figures 2, 3, 5, and 9 require clearer labeling and improved graphical quality. The current form makes interpretation challenging.

Reviewer 4 Report

Comments and Suggestions for Authors

In the manuscript “Polymorphism in Glu-Phe-Asp Proteinoids” by Drs. Panagiotis Mougkogiannis and Andrew Adamatzky authors studied how Glu-Phe-Asp assemblies how polymorphism might have led to the shift from non-living chemistry to active, responsive systems in the prebiotic space. Authors demonstrated that Glu-Phe-Asp proteinoid assemblies show a strong link between their shape and electrical properties. It was shown that microspheres create   neuron-like spikes and fibers show slow changes.  Authors concluded that heterogeneous mixtures have unique traits that hint at early information processing capabilities. These findings show new ways that prebiotic pathways can influence signal processing systems. Also, authors believe that obtained result suggest innovative methods for biomimetic computing.  By another words, simple self-assembling proteinoids can create different electrical behaviors. 

Very interesting manuscript, I have no objections to the concept, but there are some questions and comments that could improve the manuscript.

The generation of a spike-type discharge described by the authors is of a probabilistic nature. This is a cardinal difference from the generation of a spike in living cells, where the launch of a certain cascade of molecular interactions almost inevitably leads to membrane depolarization.

Are there any factors that can reduce or increase the probability of spike generation in the system described by the authors, without destroying this system?

If so, could these factors underlie the evolution of taxis - chemotaxis, thermotaxis, response to light - and, what is not less interesting - to radioactive radiation (Rowe et al, doi: 10.1016/j.plrev.2025.02.002)?

Bacteria using radiolysis have been described (Atri , 2016, doi: 10.1098/rsif.2016.0459.)  , in particular Desulforudis audaxviator - is a certain parallelism with the results obtained by the authors possible?

A related question is that since the Glu-Phe-Asp system is based on passive electrochemical processes and cannot generate fast spikes, could this system be made to generate faster spikes using an external pulsed energy source? For example, a short-term application of radiation?

The opinion of the authors and comparison of their results with these and possibly some other publications would be very important for the scientific community. The manuscript is very impressive, I will be happy to recommend the manuscript for publication after the additions have been made.

Round 2

Reviewer 1 Report

Comments and Suggestions for Authors

All the comments are properly resolved 

Comments on the Quality of English Language

Good

Reviewer 2 Report

Comments and Suggestions for Authors

The author has addressed the majority of the comments, with a few reserved for future work.

Reviewer 3 Report

Comments and Suggestions for Authors

The authors have responded to all prior critiques. 
The additional experiments and related data analysis, including the use of statistics, have been appropriately performed and the conclusions are sound. The manuscript is well-written and organized, and the data and discussion of the results clearly presented. 
The manuscript is suitable for publication and represents an innovative and important contribution to the field. No further concerns.